# Discovery of a peripheral 5HT$_{2A}$ antagonist as a clinical candidate for metabolic dysfunction-associated steatohepatitis

Haushabhau S. Pagire[1,2,11], Suvarna H. Pagire[1,2,11], Byung-kwan Jeong[3,11], Won-Il Choi[3,11], Chang Joo Oh[4], Chae Won Lim[5], Minhee Kim[1], Jihyeon Yoon[1], Seong Soon Kim[6], Myung Ae Bae[6], Jae-Han Jeon[4,7], Sungmin Song[2], Hee Jong Lee[2], Eun Young Lee[2], Peter C. Goughnour[2], Dooseop Kim[2], In-Kyu Lee[4,8], Rohit Loomba[9], Hail Kim[3,10] ✉ & Jin Hee Ahn[1,2] ✉

Metabolic Dysfunction-Associated Steatotic Liver Disease (MASLD) is currently the leading cause of chronic liver disease worldwide. Metabolic Dysfunction-Associated Steatohepatitis (MASH), an advanced form of MASLD, can progress to liver fibrosis, cirrhosis, and hepatocellular carcinoma. Based on recent findings by our team that liver 5HT$_{2A}$ knockout male mice suppressed steatosis and reduced fibrosis-related gene expression, we developed a peripheral 5HT$_{2A}$ antagonist, compound **11c** for MASH. It shows good in vitro activity, stability, and in vivo pharmacokinetics (PK) in rats and dogs. Compound **11c** also shows good in vivo efficacy in a diet-induced obesity (DIO) male mice model and in a choline-deficient, L-amino acid-defined, high-fat diet (CDAHFD) male mice model, effectively improving histologic features of MASH and fibrosis. According to the tissue distribution study using [$^{14}$C]-labeled **11c**, the compound was determined to be a peripheral 5HT$_{2A}$ antagonist. Collectively, first-in-class compound **11c** shows promise as a therapeutic agent for the treatment of MASLD and MASH.

Metabolic Dysfunction-Associated Steatotic Liver Disease (MASLD), formerly known as Nonalcoholic fatty liver disease (NAFLD) is emerging as a leading chronic liver disease worldwide[1–3]. MASLD is an increasingly common condition defined by the presence of ≥ 5% liver steatosis in the absence of significant alcohol consumption, steatogenic medication, or monogenic hereditary disorders[4]. Both Metabolic Dysfunction-Associated Steatohepatitis (MASH, formerly known as nonalcoholic steatohepatitis (NASH)) and non-alcoholic fatty liver (NAFL), known as steatosis, are histologically categorized under the umbrella of Metabolic Dysfunction-Associated Steatotic Liver

[1]Department of Chemistry, Gwangju Institute of Science and Technology, Gwangju 61005, Republic of Korea. [2]JD Bioscience Inc., TJS Knowledge Industrial Center Suite 801, 208 Beon-gil Cheomdangwagi-ro, Buk-gu, Gwangju 61011, Republic of Korea. [3]Graduate School of Medical Science and Engineering, Korea Advanced Institute of Science and Technology, Daejeon 34141, Republic of Korea. [4]Research Institute of Aging and Metabolism, Kyungpook National University School of Medicine, Daegu 41404, Republic of Korea. [5]Leading-edge Research Center for Drug Discovery and Development for Diabetes and Metabolic Disease, Kyungpook National University Hospital, Daegu 41404, Republic of Korea. [6]Bio & Drug Discovery Division, Korea Research Institute of Chemical Technology, Daejeon 34114, Korea. [7]Department of Internal Medicine, School of Medicine, Kyungpook National University, Kyungpook National University Chilgok Hospital, Daegu 41404, Republic of Korea. [8]Department of Internal Medicine, School of Medicine, Kyungpook National University, Kyungpook National University Hospital, Daegu 41944, Republic of Korea. [9]NAFLD Research Center, Division of Gastroenterology and Hepatology, Department of Medicine, University of California at San Diego, La Jolla, CA 92093, USA. [10]Biomedical Research Center, Korea Advanced Institute of Science and Technology (KAIST), Daejeon 34141, Republic of Korea. [11]These authors contributed equally: Haushabhau S. Pagire, Suvarna H. Pagire, Byung-kwan Jeong, Won-Il Choi. ✉e-mail: hailkim@kaist.edu; jhahn@gist.ac.kr

Disease (MASLD), formerly known as non-alcoholic fatty liver disease (NAFLD). Steatosis, also called simple fatty liver, is defined as the presence of a fatty liver with little to no evidence of inflammation or damage, whereas MASH is defined as the presence of fatty liver and inflammation with evidence of ballooning, with or without peri-sinusoidal fibrosis[5]. MASH can progress to more advanced stages of liver disease including cirrhosis, liver failure, and hepatocellular carcinoma (HCC)[6,7]. Obesity and type-2 diabetes are associated with elevated risk of developing MASH[8].

Serotonin (5-hydroxytryptamine (5HT)) is a monoamine neurotransmitter that modulates central and peripheral functions. Peripheral 5HT is emerging as a key regulator of systemic energy metabolism that modulates various physiological roles in multiple metabolic tissues[9–13]. As 5HT cannot cross the blood brain barrier, central and peripheral 5HT systems function separately[14]. Interestingly, among the 5HT receptors found to be expressed at detectable levels in the liver, $5HT_{2A}$ (also known as HTR2A) expression was increased by high-fat diet (HFD) feeding[15]. Our team reported that HFD-fed liver specific $5HT_{2A}$ knock-out ($5HT_{2A}$ LKO) mice showed decreased hepatic steatosis as examined by histology, NAFLD Activity Score (NAS), and hepatic triglycerides (TG) concentrations without affecting the body weight, glucose tolerance, insulin sensitivity, and plasma lipid profiles[15]. Also, inflammation and fibrosis-related gene expressions were decreased in the liver of $5HT_{2A}$ LKO mice, indicating that $5HT_{2A}$ signaling is important for inflammation and fibrosis in the liver[15]. Thus, $5HT_{2A}$ is a potential target for therapies aiming to prevent the progression of MASLD and MASH. Alarmingly, approximately 25% of MASH patients progress to cirrhosis, a serious premalignant condition that may require liver transplantation or result in liver failure[16]. Obesity-associated MASH is currently the third leading cause for liver transplantation and is expected to surpass hepatitis C as the principal cause for liver transplantation in the developed world[17]. The global prevalence of MASLD is estimated to be 25%[2] and accounts for 75.1% of all chronic liver disease. The most effective treatments currently for management of MASH are life-style interventions. Healthy diet, physical exercise, and weight loss are advised for MASH patients with an aim to halt disease progression and improve liver function[18].

There are currently no approved pharmaceutical therapies for the treatment of MASH. Off-label drugs and functional nutrients use is the only available treatment option. Pharmaceutical therapies could be an important addition and diverse drug candidates are currently in clinical trials. Obeticholic acid (OCA), commercially known as Ocaliva significantly improved hepatic fibrosis and hepatic steatosis, but these therapeutic effects did not outweigh its risk of side effects in clinical trials, and Selonsertib failed to show significantly improved hepatic fibrosis in phase III clinical trial[19]. We have been working on developing new peripheral $5HT_{2A}$ antagonists and reported amino-acid containing derivatives for MASLD. The compounds showed good in vitro activities but could not be further developed due to safety issues and deliverability as an oral drug candidate[20,21]. Therefore, we attempted to identify a new peripheral $5HT_{2A}$ antagonist suitable for oral treatment of MASH[22]. Herein, we report the synthesis and biological evolutions of a novel clinical candidate compound **11c** for MASLD or MASH treatment. Library screening and molecular docking led to the identification of compound **11c** as a potent $5HT_{2A}$ antagonist. In vitro activity, selectivity, stability, toxicity, and in vivo PK in rats and dogs were conducted for evaluation of druggability and DMPK. The tissue distribution study using [$^{14}$C]-labeled **11c** was performed to determine whether to be a peripheral agent. In vivo efficacy in diet-induced obesity (DIO) mice model and in a choline-deficient, L-amino acid-defined, high-fat diet (CDAHFD) mice model, was evaluated for MASLD or MASH.

## Results
### Discovery of a potent $5HT_{2A}$ antagonist
Based on previous $5HT_{2A}$ LKO results by our team[15], we have been synthesizing new (peripheral) $5HT_{2A}$ antagonists. In this paper, we tried to identify new hits through the drug library and/or rational design from FDA approved neuronal drugs. Therefore, we searched for a new scaffold and found compound **2** that is known as Clarinex (Desloratadine), which is used to treat allergies (Supplementary Fig. 1). Compound **2** is a tricyclic compound with a peripheral $H_1$-antagonist action, was identified as a hit with $IC_{50}$ value of 232 nM for $5HT_{2A}$. According to the initial finding, we attempted to develop a new peripheral $5HT_{2A}$ antagonist. Based on molecular docking support, diverse derivatives were synthesized (Supplementary Figs. 2, 3), and the inhibitory potency of the synthesized compounds towards $5HT_{2A}$ was evaluated, and the results were summarized in Tables 1−4. Therefore, we decided to explore different substituents on nitrogen of piperidine moiety of compound **2** by introducing N-acetyl (**3**), N-methylsulfonyl (**4**), isopropyl thiourea (**5**) and methyl (**6**) groups. Compounds **3-5** (Table 1) showed significantly decrease in activity compared to compound **2**, on the other hand the methyl substituent derivative **6** showed improved activity ($IC_{50}$ = 119 nM).

To further expand the substituents on N-methyl piperidine derivative series, more substituted derivatives **8a−c** were synthesized and evaluated as shown in Table 2. As compared to derivative **6, 8a** displayed improved activity with an $IC_{50}$ value of 81.37 nM. **8b** and **8c**, which have bicyclic moieties, showed weaker or similar percent inhibition activities (68% and 73% inhibition at 1 uM) Next, we examined whether the reduction of keto to alcohol could further improve the activity. Thus, compounds **9a−c** were synthesized as shown in Table 2, and evaluated for their activity. They showed good in vitro activities, but all alcohol derivatives exhibited low liver microsomal stability.

Based on the structure and activity of the two series above, we replaced the alcohol with −$CH_2$. As shown in Table 3, the inhibitory activity of compounds **11a−d** containing the ethyl linker were similar to better than that of compounds **9a−c** with 2-ethanol linker. Compounds **11b, 11c,** and **11d** were the most active compounds with an $IC_{50}$ value of 14, 14 and 47 nM, respectively. Compound **11c** is metabolically stable in buffer as well as in plasma (Table 5), but compound **11b** and **11d** were metabolically unstable (11b = 11.7% and 17.3%, 11d = 0.50% and 5.33% in Rat and Human respectively) after 30 min incubation in the liver microsome. Additionally, we attempted to synthesize compound **11c** derivatives without a substituent (**11e**) and with a methyl (**21**) substituent on the tricyclic core instead of chloro. Compounds **11e** and **21** showed less potency than compound **11c**. Moreover, we separated two enantiomers (**22 f** and **22 g**) of compound **11c** by using the reported synthesis procedure[23]. Both enantiomers (+)-**22f** ($IC_{50}$ = 10.6 nM) and (-)-**22g** ($IC_{50}$ = 11.08 nM) showed similar activity and drug metabolism and pharmacokinetics (DMPK) profiles therefore, we chose racemic compound **11c** for further biological evaluation.

### Molecular docking
To better optimize compound **2** (hit), molecular docking was performed using the crystal structure of human $5HT_{2A}$ (Protein Data Bank 7WC4). First, docking of compound **2** was performed (Fig. 1a, left panel) and seven interactions were determined (Fig. 1a, right panel). Based on these interactions, we introduced diverse moieties but found that the nitrogen group derivatives were critical for efficacy. The modeling was analyzed, and the nitrogen of compound **2** showed pocket-binding interaction with serine 242, SER-242. Docking of compound **11c** (Fig. 2b, left panel) had an increased number of interactions to ten for compound **11c** (Fig. 2b, right panel). Compound **11c** bound to all the same side chains as compound **2** (ILE205, PRO-209, ILE-210, PHE-234, ILE-237, VAL-241, SER-242) with the addition of three more interactions (TRP-164, SER-203, ILE-202). In addition, we performed docking of $5HT_{2A}$ antagonists, Risperidone, and Zotepine that was previously reported[24]. We compared binding affinities and it was shown that compound **11c** had a better binding affinity than Compound 2 and commercially available Resperidone and Zoptepine, (−9.2, −8.0, −8.4, and −6.3 kcal/mol respectively). Risperidone showed

**Table 1 | SAR Exploration of compounds 1-6**

| Compound | Structure | % Inhibition at 10 µM (SD) | IC$_{50}$ (nM) |
|---|---|---|---|
| **1** | | 2 | ND |
| **2** | | 88 ± 2.60 | 232 |
| **3** | | -1 | ND |
| **4** | | 2 | ND |
| **5** | | 8 | ND |
| **6** | | 95 ± 0.31 | 119 |

*SD* Standard Deviation, *ND* Not determined.

**Table 2 | SAR Exploration of compounds 8a-c and 9a-c**

| Compound | Structure | % Inhibition at 1 µM (SD) | IC$_{50}$ (nM) |
|---|---|---|---|
| **6** | | 90 ± 0.62 | 119 |
| **8a** | | 83.2 ± 1.98 | 81.37 |
| **8b** | | 68.34 ± 2.78 | ND |
| **8c** | | 73.02 ± 0.51 | ND |
| **9a** | | 92 ± 1.07 | 31.95 |
| **9b** | | 92.18 ± 1.59 | 21.35 |
| **9c** | | 93.85 ± 3.13 | 22.19 |

*SD* Standard Deviation, *ND* Not determined.

better binding than Zotepine in our molecular docking, which could correlate to better efficacy that was previously reported[25]. This analysis further confirms that compound **11c** ability as an antagonist for 5HT$_{2A}$.

### DMPK, selectivity and toxicity

Pharmacokinetic parameters of compound **11c** were determined and summarized in Table 4. Compound **11c** demonstrated good PK profiles with the half-life was determined to be 4.14 h and 8.6 h in rats and dogs respectively for intravenous administration, and 10.2 h in dogs for oral administration while its bioavailability was 63% (5 mpk, rat), 92.5% (10 mpk, rat) and 73% (5 mpk) in dogs respectively. Compound **11c** also exhibited a reasonable area under the curve (AUC, Table 4).

Compound **11c** showed good chemical, hepatocyte, and plasma stability, no significant cytotoxicity in five representative mammalian cell lines (VERO, HFL-1, L929, NIH3T3, CHO-K1) and low CYP inhibition in five subtypes (Table 5). Compound **11c** didn't show mutagenic potential (AMES test) and exhibited LD$_{50}$ value of over 1000 mpk in acute tox study. For further evaluation, the inhibition percentages of other subtypes of serotonin receptors were screened and compound **11c** showed selective inhibition of 5HT$_{2A}$ among the subtypes. Although compound **11c** also inhibited 5HT$_{2c}$ and 5HT$_7$, they are mainly expressed in brain[26,27] therefore peripheral compound **11c** may not inhibit them.

Using radio labeled [$^{14}$C]-compound **11c**, we conducted the tissue distribution study and the results are summarized in Table 6. [$^{14}$C]-

**Table 3 | SAR Exploration of compounds 11a-e, 21 and 22f-g**

| Compound | Structure | % Inhibition at 1 µM (SD) | IC$_{50}$ (nM) |
|---|---|---|---|
| 9c | | 93.85 ± 3.13 | 22.19 |
| 11a | | 94.47 ± 1.53 | ND |
| 11b | | 92.63 ± 0.60 | 14 |
| 11c | | 92.56 ± 0.77 | 14 |
| 11d | | 102 ± 0.03 | 47 |
| 11e | | 89.22 ± 10.98 | 61.01 |
| 21 | | 28.80 ± 3.68 | 67.74 |
| 22 f | | 96.07 ± 1.87 | 10.6 |
| 22 g | | 97.17 ± 7.11 | 11.08 |

*SD* Standard Deviation, *ND* Not determined.

compound **11c** was below the lower limit of quantitation in the brain, and it remained mainly in liver, intestine, and pancreas.

## In vivo efficacy of compound 11c in DIO mice

It has been reported that genetic deletion or inhibition of 5HT$_{2A}$ has an anti-MASLD through the reduction of lipid accumulation or inhibition of de novo lipogenesis in the liver[15,28]. Accordingly, we evaluated the potential in vivo efficacy of compound **11c** for MASLD. C57BL6/J mice were fed a HFD for eight weeks and were administered 5 or 10 mpk of compound **11c** daily oral gavage. The body weight gain upon HFD was attenuated and fat mass was reduced in mice treated with compound **11c** (10 mpk) compared to vehicle-treated mice (Fig. 3a, b). Glucose tolerance was improved by compound **11c** (5 or 10 mpk) treatment (Fig. 3c). Tissue weight of liver and inguinal white adipose tissue (iWAT) significantly decreased in compound **11c**-treated mice (Fig. 3d). In the histological analysis using H&E staining, steatosis, lobular inflammation, and hepatocyte ballooning in liver tissue was markedly reduced in compound **11c**-treated mice in a dose-dependent manner (Fig. 3e, f). Furthermore, compound **11c**-treated mice showed decreased hepatic TG concentration (Fig. 3g), and the expression of lipogenesis-related genes were reduced in compound **11c**-treated mice (Fig. 3h). These data indicated that compound **11c** could protect against diet-induced MASLD.

## In vivo efficacy of Compound 11c for liver fibrosis and inflammation

Our previous report revealed that genetic deletion of 5HT$_{2A}$ resulted in the downregulation of genes related to hepatic fibrosis and inflammation[15]. To determine the in vivo efficacy of compound **11c** for liver fibrosis and inflammation, the mice were subjected to a choline-deficient, L-amino acid-defined high-fat diet (CDAHFD) for 12 weeks and administered compound **11c** daily by oral gavage. During the study period, no abnormal clinical signs were observed, and no significant difference in food intake was noted amongst CDAHFD-fed groups (Supplementary Fig. 4). Body weights of mice were significantly decreased in CDAHFD-fed groups compared to the normal diet-fed group (vehicle) but the body weight changes among the CDAHFD groups were not significantly different (Fig. 4a). Collagen accumulation and fibrosis were assessed by examining the levels of Col1a1 and α-SMA expressions in liver tissue. The mRNA expressions of both ol1a1 and α-SMA in liver tissue were significantly increased in CDAHFD-fed vehicle group compared to control and were significantly decreased in CDAHFD-fed **11c** and compared to CDAHFD-fed vehicle group (Figs. 4b, 3c). Similarly, collagen accumulation assessed in the liver tissue using Sirius red staining showed a substantial increase in the CDAHFD-fed vehicle group comparing normal-diet group and a significant decrease in CDAHFD-fed compound **11c** and compared to CDAHFD-fed vehicle group (Fig. 4d, e). Immunohistochemical staining of liver tissue was conducted to assess the expression of α-SMA, a marker of fibrosis, TNF-α, and IL-1β, a maker of inflammation. It was shown that α-SMA, TNF-α, and IL-1β expressions were significantly increased in the CDAHFD-fed vehicle group compared to normal diet-fed group, and a significant decrease was observed in the group treated with compound **11c** compared to the CDAHFD-fed vehicle group (Fig. 4f–h). Overall, the efficacy of compound **11c** was evaluated in an in vivo mouse model of liver fibrosis and inflammation following a CDAHFD for 12 weeks. Compound **11c** administration at 5 and 10 mpk dose for 12 weeks had significant reduction in the RNA expressions of Col1a1 and α-SMA were observed in the liver of the mice treated with compound **11c** administration at both doses compared to the CDAHFD-fed vehicle control animals. These findings clearly indicate that compound **11c** has a therapeutic benefit for the treatment of liver fibrosis and inflammation with liver disease. These beneficial effects of compound **11c** on hepatic inflammation and fibrosis could be attributed to the reduction in hepatic lipid accumulation. However, considering that elevated ROS production and mitochondrial

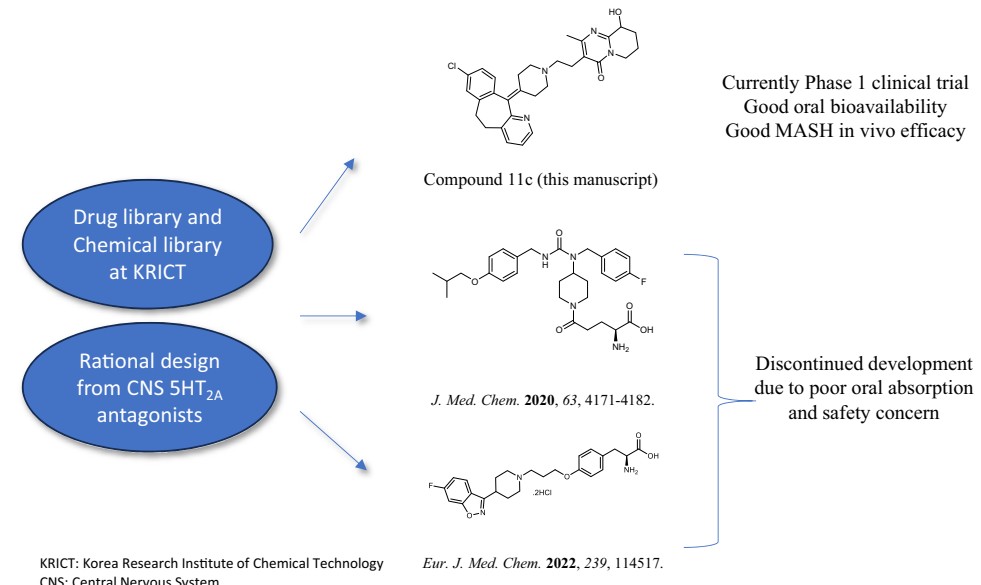

Currently Phase 1 clinical trial
Good oral bioavailability
Good MASH in vivo efficacy

Compound 11c (this manuscript)

*J. Med. Chem.* **2020**, *63*, 4171-4182.

Discontinued development
due to poor oral absorption
and safety concern

KRICT: Korea Research Institute of Chemical Technology
CNS: Central Nervous System

*Eur. J. Med. Chem.* **2022**, *239*, 114517.

**Fig. 1 | Strategy and history of 5HT$_{2A}$ antagonists.** Library and rational design for the development of compound **11c** as a potent 5HT$_{2A}$ antagonist. Previous research efforts were discontinued due to limited oral absorption and safety.

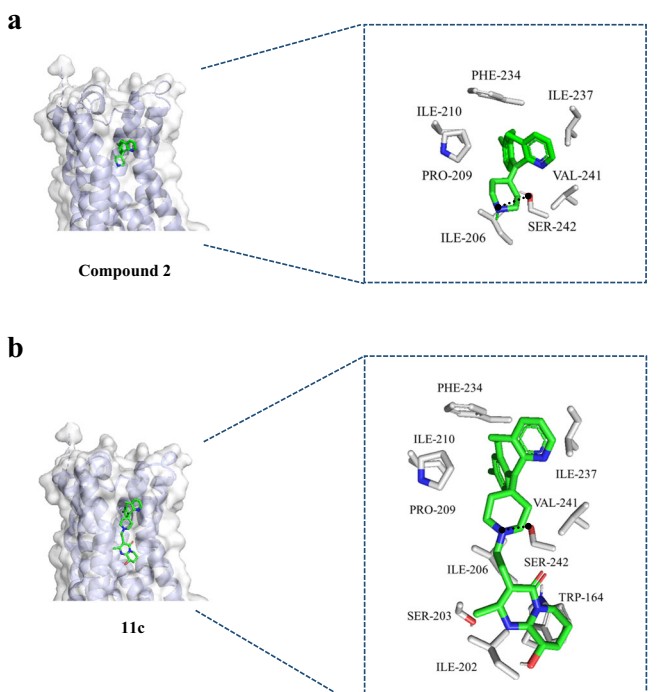

**a**

Compound 2

**b**

11c

**Fig. 2 | Molecular docking of compounds to Human 5HT$_{2A}$. a** The predicted binding model of compound **2** binding to 5HT$_{2A}$ (left panel) and interactions with 5HT$_{2A}$ side chains (right panel). **b** The predicted binding model of compound **11c** binding to 5HT$_{2A}$ (left panel) and interactions with 5HT$_{2A}$ side chains. The three-dimensional structure of human 5HT$_{2A}$ is shown with a ribbon expression. 5HT$_{2A}$ side chains and compounds **2** and **11c** are represented as sticks. *PHD* phenylalanine, *ILE* Isoleucine, *PRO* Proline, *VAL* valine, *SER* Serine, *TRP* Tryptophan.

stress are important in the development of hepatic inflammation and fibrosis in CDAHFD mouse model, our data also suggested that anti-inflammation and fibrosis effects of compound **11c** could be attributed to the inhibition of 5HT$_{2A}$ signaling in other target cells such as macrophage and hepatic stellate cells.

## Discussion

In this study, we developed compound **11c** as a novel peripheral 5HT$_{2A}$ antagonist for MASH and steatosis. Currently there is no approved treatment for MASH, although diverse drug candidates are undergoing clinical trials. Our goal is to identify oral and non-BBB permeable 5HT$_{2A}$ antagonists for chronic disease including MASLD and MASH. We discovered new hits by drug library screening and employing rational design. Fortunately, Desloratadine, which is known as a peripheral agent, was identified as a hit with IC$_{50}$ value of 232 nM. Molecular docking supported the optimization process to identify compound **11c**, which showed good in vitro activity with an IC$_{50}$ value of 14 nM, using the crystal structure of human 5HT$_{2A}$. Compound **11c** is a racemic mixture, therefore, both chiral isomers were synthesized and evaluated for their in vitro activities. Both isomers showed the same activities, so racemic **11c** was used for further evaluation. Compound **11c** exhibited a good chemical profile by having hepatocyte and plasma stability with no significant cytotoxicity and low CYP inhibition. Compound **11c** didn't show mutagenic potential (AMES test) and exhibited LD$_{50}$ value of over 1000 mpk in the acute tox study. In PK profiles, it had over 60% oral bioavailability in rats and dogs. According to the tissue distribution study with [$^{14}$C]-labeled compound **11c**, it was determined to be a peripheral 5HT$_{2A}$ antagonist. Compound **11c** demonstrated significant efficacy in reducing fatty liver and fibrosis in both DIO and CDAHFD mice models.

We also showed the possibility of the direct anti-inflammatory and anti-fibrotic effects of compound **11c** using CDAHFD mouse model. Although we did not show any direct evidence of 5-HT's role in inflammation and fibrosis, 5-HT is known to regulate the activation and proliferation of hepatic stellate cells (HSCs) and macrophages[29–31]. Thus, further studies are being conducted to elucidate the detailed mechanisms of how compound **11c** can reduce hepatic inflammation and fibrosis.

In conclusion, we suggest that compound **11c** is a promising drug candidate, which functions as a peripheral 5HT$_{2A}$ antagonist aiming for MASLD and liver fibrosis, which is in phase 1 clinical trial (NCT05517564)[32].

**Table 4 | Pharmacokinetic Study of Compound 11c in rat and dog models**

| Parameters | In vivo PK (rat) | | | In vivo PK (dog) | |
|---|---|---|---|---|---|
| | IV (5 mpk) | PO (5 mpk) | PO (10 mpk) | IV (5 mpk) | PO (5 mpk) |
| $T_{1/2}$ (h) | 4.4 | 2.5 | 2.7 | 8.6 | 10.2 |
| AUC (µg·h/mL) | 1.55 | 0.98 | 2.88 | 16.79 | 19.11 |
| CL (L/h/kg) | 2.82 | - | - | 0.42 | - |
| V (L/kg) | 8.86 | - | - | 4.36 | - |
| F (%) | - | 63 | 92.5 | - | 73 |

*(T1/2)* Half-life, *AUC* Area under the curve, *CL* Clearance, *V* Volume of Distribution, *F* Bioavailability, *PO* per so, by mouth), *IV* Intravenous, *mg/kg* (mpk).

**Table 5 | Stability, cytotoxicity, CYP, AMES, solubility, acute toxicity, and selectivity results of compound 11c**

| Assay | Result |
|---|---|
| Chemical stability | 99.8% ($25^0$C, 3weeks), 99.6% ($60^0$C, 3weeks) |
| Hepatocyte stability (% remaining after 3 h) | 94%(human), 88.5% (dog), 84.6% (monkey), 68.9% (rat), 75% (mouse) |
| Plasma stability (% remaining after 4 h) | >99.9% (m), >99.9% (r), >98.6 + 11.7% (h), |
| CYP inhibition (10 µM) | 1A2: < 1%, 2C9:3.17% 2C19:6.83% 2D6: 26.3% 3A4:17.6% |
| Cytotoxicity ($IC_{50}$) | VERO > 100 µM, HFL-1 > 100 µM, L929 > 100 µM, NIH3T3 > 100 µM, CHO-K1 > 100 µM |
| AMES | Negative |
| Solubility | 1.3 mg/mL pH = 6.8 |
| Acute toxicity | $LD_{50}$ > 1000 mpk |
| Selectivity | 1 A: 9 % inhibition 1B: 16 % inhibition 1D: 3 % inhibition 2 A: 102 % inhibition, $IC_{50}$ = 14 nM 2B: 94 % inhibition, $IC_{50}$ = 290 nM 2 C: 103 % inhibition, $IC_{50}$ = 170 nM 3: -1 % inhibition 4E: -3 % inhibition 6: 7 % inhibition 7: 91 % inhibition, $IC_{50}$ = 100 nM |

## Methods

### Animal experiments

For in vivo efficacy in DIO models and CDAHFD models, the experimental protocol for this study was approved by the Institutional Animal Care and Use Committee (IACUC) at the Korea Advanced Institute of Science and Technology (No. KA2023-027-v2) or Kyungpook National University (No. KNU-20200134). All relevant ethical guidelines were followed. C57BL/6 J male mice (11 weeks old) were purchased from the Charles River Japan and were housed in a specific pathogen-free barrier facility under a regular light-dark cycle (12-hour light/12-hour dark) at 24°C with 40–60% humidity. Food (standard chow diet, Envigo, 2018S) and water were provided ad libitum. At 12 weeks, mice were fed a high-fat diet (HFD, D12492, Research Diet) with vehicle or compound **11c** treatment daily by oral gavage. Mice were euthanized by the method of cervical dislocation. The measurement of lean and fat mass in live mice was performed by a body composition analyzer (LF50 BCA-analyzer, Bruker). After 6 h of fasting, mice were sacrificed, and tissue samples were obtained.

**Table 6 | Pharmacokinetic parameters of compound 11c**

| Tissue | Cmax (ng equiv./g) | Tmax (h) | $T_{1/2}$ (h) | AUC0-t (ng equiv./g) | AUCinf (ng equiv./g) | Tissue: plasma** |
|---|---|---|---|---|---|---|
| Plasma | 225 | 0.5 | NC | 783 | NC | 1.00 |
| Whole Blood | 158 | 0.5 | NC | 589 | NC | 0.75 |
| Adrenal gland | 1510 | 6 | NC | 7210 | NC | 9.21 |
| Brain | BQL | NC | NC | NR | NC | NA |
| Heart | 369 | 6 | NC | 1800 | NC | 2.30 |
| Kidney | 1840 | 6 | NC | 7860 | NC | 10.0 |
| Large Intestine | 9800 | 6 | NC | 127000 | NC | 162 |
| Liver | 2920 | 6 | NC | 15400 | NC | 19.7 |
| Lung | 941 | 6 | NC | 4290 | NC | 5.48 |
| Pancreas | 4330 | 6 | NC | 15400 | NC | 19.7 |
| Small Intestine | 45000 | 0.25 | NC | 16600 | NC | 21.2 |
| Spleen | 2290 | 6 | NC | 9520 | NC | 12.2 |
| Stomach | 3240 | 0.25 | NC | 5130 | NC | 6.55 |

Equivalents in Plasma and Tissue of Male Sprague Dawley Rats following an Oral Dose of [$^{14}$C]-11c at 5 mg/kg. (NC= Not calculated due to insufficient terminal elimination phase, *NA* Not applicable, *NR* Not reportable due to <3 consecutive quantifiable concentrations, *BQL* Below the lower limit of quantitation 148 ng equiv./g, **Tissue: plasma ratio based on AUC0-t).

### Pharmacokinetic study in rats and dogs

The experimental protocol for this study was approved by the IACUC of KPC Co., Ltd. (No. KPC IACUC-P203046; Beagle dogs) or QuBEST BIO (No.QBIACUC-A20131). All relevant ethical guidelines were followed. Beagle male dogs (11 ~ 20 months old) were purchased from ORIENT BIO Inc. and were housed in the barrier animal facility of KPC Co., Ltd. under a regular light-dark cycle (12-hour light/12-hour dark) at 21.1–23.4°C with 55.3–72.2% humidity. Food (standard irradiated pelleted commercial laboratory diet, Cargill Agri Purina, 300 g/dog.day) and water were provided ad libitum. All study dogs were returned to the laboratory colony after the study without any euthanasia. Sprague-Dawley male rats (7 weeks old) were purchased from SAMTAKOBIO-KOREA, Inc. and were housed in the barrier animal facility of Nonclinical Research Center, QuBEST BIO under a regular light-dark cycle (12-hour light/12-hour dark) at 19.7–22.7°C with 54.5–65.6% humidity. Food (standard irradiated pelleted commercial laboratory diet, Purina Rodent Chow, 38057) and water were provided ad libitum. All study rats were euthanized by $CO_2$ gas upon study completion.

### Chemistry

Detailed synthesis protocols and characterization of compounds can be found in the Supplementary Methods section of the Supplementary Information.

### Human 5HT$_{2A}$ inhibition assay

The derivatives were evaluated at a concentration of 1 µM for their antagonist activity on the 5HT$_{2A}$ receptor (Tables 1–3)[33,34]. Human 5HT$_{2A}$ stably transfected HEK293 cells (a type of cell derived from kidney embryos)[20] were seeded in a 96-well plate (Costar; 3603) at a density of $9 \times 10^4$ in 100 µL of minimum essential medium (Welgene; LM007-86) containing 10% dialyzed fetal bovine serum (Gibco; 30067-334) and 1% penicillin/streptomycin (Gibco; 15140-122) and kept in a 37 °C incubator for 24 h. The cells were stained with 100 µL of cytosolic calcium dye using the FLIPR calcium 6 assay solution (Molecular devices, #R8190) supplemented with 2.5 mM probenecid (Sigma; P8761) and incubated for 1 h at 37 °C. After 1 h, the designated amount of the compound was treated in each well and incubated for another 1 h at 37 °C. Then, the cells were loaded in a

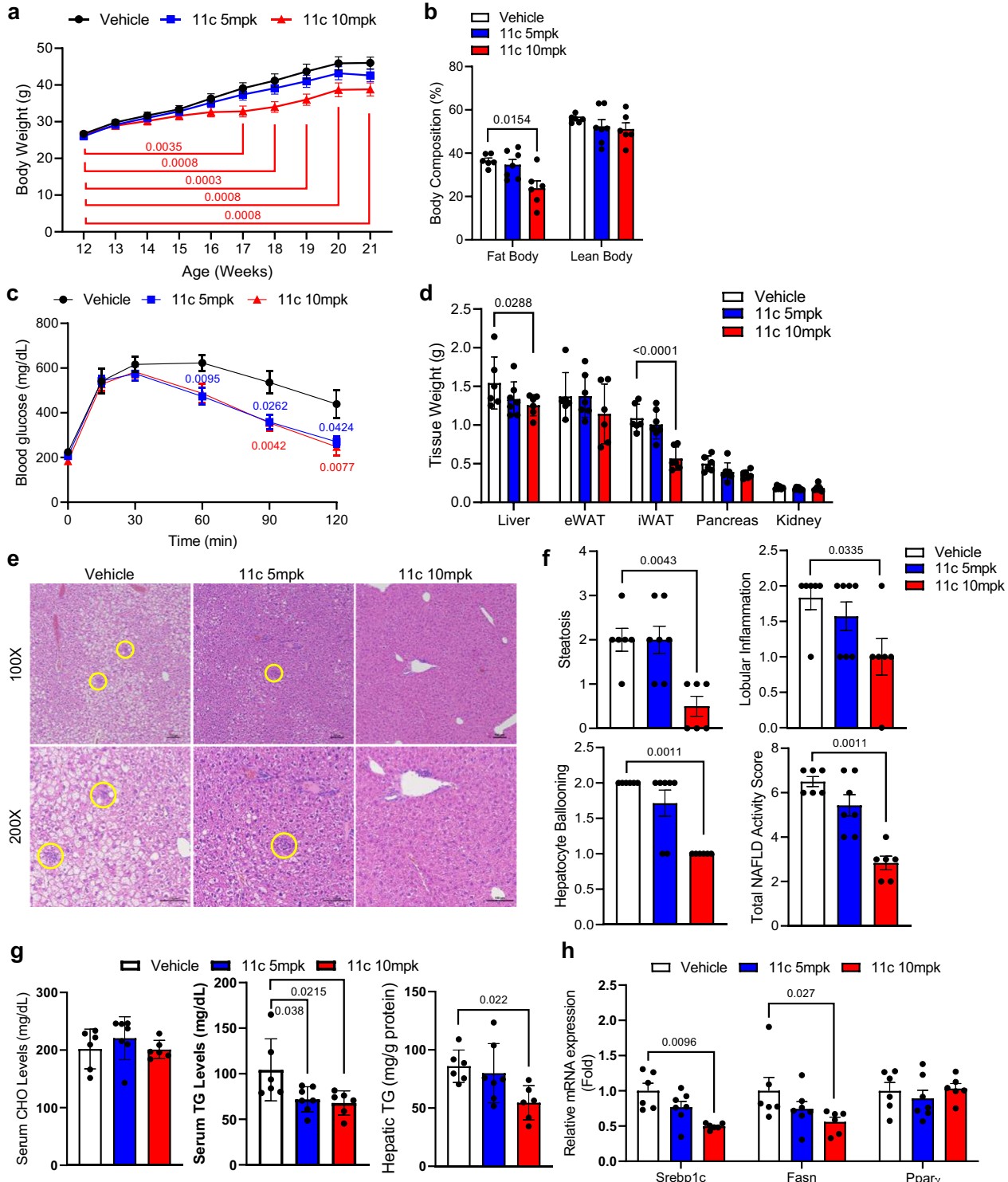

**Fig. 3 | 5HT$_{2A}$ antagonist, compound 11c, attenuates HFD-induced obesity, MASLD, and metabolic syndrome. a–h** Twelve-week-old mice were treated with vehicle or compound **11c** (5 and 10 mpk) daily oral gavage with a high-fat diet for eight weeks, $n = 6$ for vehicle, $n = 7$ for compound 11c 5mpk, and $n = 6$ for 10mpk. **a** body weight trend of vehicle- or compound **11c**-treated mice, two-way ANOVA with *post hoc* Dunnett's test. **b** Percent fat and lean mass (body composition) of vehicle- or compound **11c**-treated mice, two-way ANOVA with *post hoc* Dunnett's test. **c** Intraperitoneal glucose tolerance test (IPGTT) after 16 h fasting, two-way ANOVA with *post hoc* Dunnett's test. **d** Tissue weight of vehicle-treated or compound **11c**-treated mice, two-way ANOVA with *post hoc* Dunnett's test.

**e** Representative liver histology by H&E staining in vehicle- or compound **11c**-treated mice. Scale bars, 100 μm. **f** Total NAFLD activity score (NAS) of vehicle- or compound **11c**-treated mice. Scale bars, 100 μm, 2-sided Mann-Whitney's U-test. **g** Serum cholesterol (CHO), Serum triglyceride (TG), and hepatic TG levels of vehicle- or compound **11c**-treated mice, one-way ANOVA with *post hoc* Dunnett's test. **h** Relative mRNA expression of lipogenesis-related genes in vehicle- or compound **11c**-treated mice, two-way ANOVA with *post hoc* Tukey's Honestly Significant Difference (HSD) test. Data are expressed as the means Data are expressed as the means ± SEM.

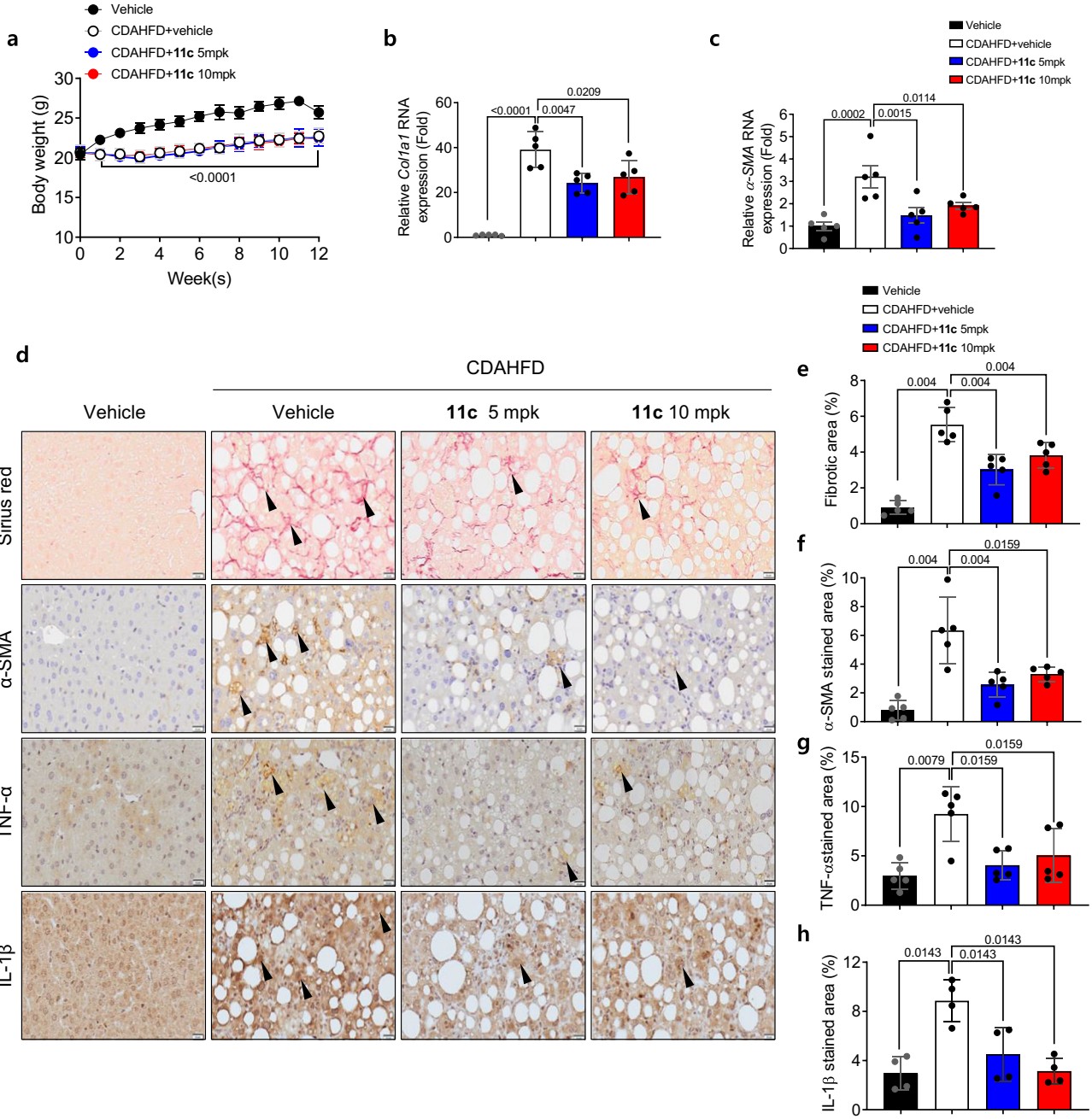

**Fig. 4 | In vivo efficacy of Compound 11c for CDAHFD-induced liver fibrosis and inflammation C57BL/6 J mice (6 weeks old) underwent 12 weeks of CDAHFD feeding and for the same period oral injections of compound 11c. a** Body weight of CDAHFD-fed control and compound **11c**-treated mice, one-way ANOVA with *post hoc* Tukey's Honestly Significant Difference (HSD) test. The data are presented as mean ± SD (*n* = 5 per group). **b, c** Relative liver *Col1a1* and *α-SMA* RNA expressions in mice, measured using quantitative RT-PCR, one-way ANOVA with *post hoc* Tukey's Honestly Significant Difference (HSD) test. The data are presented as mean ± SEM (*n* = 5 per group). **d** Representative sirius red and immunohistochemical staining images (α-SMA, TNF-α, and IL-1β) from CDAHFD-subjected vehicle and compound **11c**-treated mice (scale bars, 20 µm, arrows: positive areas). **e–h** Percentage of the sections positive for fibrotic area (sirius red), α-SMA, TNF-α, and IL-1β, 2-sided Mann-Whitney's *U*-test. The data are presented as mean ± SD (*n* = 4–5 per group).

Flexstation 3 microplate reader (Molecular Devices; FlexStaion) and 50 µL of 5 µM 5HT (serotonin hydrochloride, Sigma; H9523) was injected from a source plate (Nunc 96-well polystyrene conical bottom MicroWell Plate, Thermo; 249662) and the fluorescence intensity (Ex/Em-485nm/ 525 nm) was measured. (Supplementary Fig. 5) For determining the IC$_{50}$ values of other 5HT subtypes, an Inhibition assay was performed by Eurofins Cerep (Study ID: FR095-0014987, FR095-0016438). Compound **11c** was tested at several concentrations ranging from 3.0E-09 M to 1.0E-05 M, and the IC$_{50}$ value was calculated from concentration-response curves. Experimental conditions for the 5HT subtypes can be found in the Supporting Information (Supplementary Table 1).

## Hepatocyte stability test

Hepatocyte incubations for the test compound (10 µM final concentration, *n* = 2 per each time point) were carried out in mouse, rat, dog, monkey and human cryopreserved hepatocytes, resuspended at a density of 0.5 million cells/ml of modified Krebs-Henseleit buffer (KHB). The information regarding the hepatocytes used in the experiment is as follows: liverPool® 10-donor mixed gender human

(BioIVT, Cat.No. X008001, lot EAV), male cynomolgus monkey (BioIVT, Cat.No. M00305, lot COJ), male Beagle dog (BioIVT, Cat.No. M00205, lot USY), male Sprague Dawley rat (BioIVT, Cat.No. M00005, lot TQE) and male CD1 mouse (BioIVT, Cat.No.005052, lot LFF). In addition, the compound was also incubated with blank KHB. The final concentration of solvent (DMSO) was 0.1% of the final volume. The incubations were performed for 3 hours at 37°C and pH 7.4. Reactions were terminated at 0 and 180 minutes by the addition of 3 volumes of stop solution (acetonitrile/methanol (2:1, v/v) containing internal standard–diclofenac at 100 ng/ml). Samples (compound spiked buffer and compound spiked hepatocytes, as well as blank samples, i.e. hepatocytes with the same DMSO%) were then centrifuged and aliquots were transferred to 96-deep micronic tubes and frozen at -80 °C until LC-MS/MS analysis. The metabolic activity of hepatocytes was verified by simultaneous incubation of the following assaycontrols: testosterone (substrate for Phase I enzymes from CYP3A enzyme family, high clearance in all species), umbelliferone (substrate for Phase II conjugation enzymes: UDP glucuronyl transferases (UGT) and sulfotransferases (SULT); high clearance in all speciess) and caffeine (stable compound). All controls were within acceptable range in all tested species. The Quantification of compound **11c** was done on a Q-TOF X500R high resolution mass spectrometer (Sciex) equipped with a TurboIonSpray source and coupled to Shimadzu Nexera Series 30 HPLC pumps and autosampler.

## Molecular docking

The compounds were designed with Chemdraw 3D Ultra and converted to PDB files for docking and were designated as ligands. Human $5HT_{2A}$ crystal structure was obtained from Protein Data Bank (7WC4)[35] was used as a designated receptor. AutoDock Vina[36] was used to prepare the ligand and receptors for docking. Docking was performed using AutodockTools with grid box center coordinates -27.980, -10.325, 113.215 and grid dimensions 80 x 100 x 126 with an exhaustiveness of eight. The interactions of the side chains of $5HT_{2A}$ within four Angstroms was visualized with PyMOL2.5[37].

## In vivo efficacy for choline-deficient, L-amino acid-defined high-fat diet (CDAHFD)-induced liver fibrosis and inflammation

C57BL/6 J male mice were purchased from the Japan SLC. Inc (Hamamatsu, Japan). Mice (5 weeks old) were fed Teklad Certitied Irradiated Global 18% protein rodent diet purchased from Dooyeol Biotech Co. Ltd (Seoul, South Korea) during the acclimation period. The liver fibrosis/inflammation-induced groups were CDAHFD-fed (A06071302, Research diet). After the acclimation period, mice (6 weeks old) were divided into four groups (Vehicle, CDAHFD, CDAHFD treated with 5 and 10 mpk of compound **11c**, $n = 5$/group) for each experimental regime. Compound **11c** and vehicle (40 mM citrate buffer, pH 3.0) groups were orally administrated once daily for 12 weeks. All animal experiments complied with the relevant ethical regulations.

## Collagen I and α-SMA RNA expression in liver tissue

The frozen liver tissue was homogenized and put into QiAzol (Qiagen, Hilden) to isolate RNA, and 4 μg of RNA was synthesized into cDNA using the RevertAid First Strand cDNA synthesis kit (Thermo Fischer, Waltham). The expressions of mouse collagen I (Forward: 5'-TGT GTT CCC TAC TCA GCC GTG-3'. Reverse: 5'-CAT CGG TCA TGC TCT CTC CAA-3'), mouse α-SMA (Forward:5'-TCC TGA CGC TGA AGT ATC CGA T-3', Reverse: 5'-GGC CAC ACA AAG CTC GTT ATA G-3') and mouse 36B4 (Forward: 5'-ACC TCC TTC TTC CAG GCT TT-3', Reverse: 5'-CTC CAG TCT TTA TCA GCT GC-3') as house-keeping gene were measured using ViiA 7 Real-time PCR system (Thermo Fischer scientific).

## Sirius red and immunohistochemical staining

Mouse liver tissues were fixed in 4% formalin solution (Sigma–Aldrich, St. Louis, MO, USA) and embedded in paraffin. 4-µm-thick liver sections

were deparaffinized in xylene, and rehydrated through descending grades of ethanol. To evaluate liver fibrosis, sections were stained with Sirius red solution and subjected to immunohistochemical staining[38]. Tissue sections, after deparaffinization, underwent a 16-hour incubation in a solution containing 0.1% Direct Red 80 (365548, Sigma-Aldrich) dissolved in Picric acid (P-201, Spectrum Chemical, CA, USA) for Sirius red staining. The sections were briefly immersed in a 0.01 N HCl solution (1129, Ducksan, Republic of Korea) for 2 min. and then rinsed with distilled water for destaining. Subsequently, the sections underwent dehydration through descending grades of ethanol. Immunohistochemical staining was performed using the Polyview Plus HRP-DAB (anti-Rabbit) Kit (ENZ-KIT159-0150, ENZO Life Science, NY, USA) following the manufacturers' instructions. Deparaffinized sections were boiled in IHC-TekTM Epitope Retrieval solution (IW-1100, IHCWORLD, MD, USA) for 45 min., cooled for 1 h, incubated in a 3% $H_2O_2$ solution (69355-0350, JUNSEI, Japan) for 15 min., washed, and then incubated overnight with primary antibodies (α-SMA, ab5694, Abcam, 1:500 dilution; TNF-α, ab6671, Abcam, 1:100 dilution and IL-1β, ab9722, Abcam, 1:100 dilution). Subsequently, the sections were treated with Polyview Plus HRP for 2 h and developed with DAB chromogen/substrate. Stained kidney sections were confirmed by light microscopy (Olympus BX53 upright microscope, Tokyo, Japan). The degree of hepatic fibrosis and immunohistochemical staining was quantified by morphometric analysis of stained liver sections using NIH ImageJ software.

## Statistical analysis

Statistical analysis was performed using GraphPad Prism 8 (MA, USA). All data are presented as mean ± standard error of the mean (SEM) of at least three independent experiments as indicated in the figure legends. Statistical analyses were performed using one-way analysis of variance (ANOVA) or two-way ANOVA to compare more than three groups, followed by *post hoc* Dunnett's test or Tukey's Honestly Significant Difference (HSD) test, and Mann–Whitney's *U*-test used for two groups nonparametric comparison as appropriate. A value of $P < 0.05$ was considered to be statistically significant.

## Histological analysis

Epididymal, inguinal, and liver were harvested, fixed in 10% (w/v) neutral buffered formalin solution (Sigma-Aldrich), and embedded in paraffin. Five-micron-thick tissue sections were deparaffinized, rehydrated, and stained with hematoxylin and eosin (H&E)[10]. The total NAFLD activity score (NAS) is a widely used measure of grading for steatosis, lobular inflammation, and ballooning. The total NAS ranges from 0-8 and is the sum of unweighted scores for steatosis (0: <5%, 1: 5 ~ 33%, 2: 33 ~ 66%, 3: >66% fat accumulation in the liver), lobular inflammation (0: no foci inflammation in x200 field, 1: <2 foci inflammation, 2: 2 ~ 4 foci inflammation, 3: >4 foci inflammation), and hepatocellular ballooning (0: none, 1: few ballooning cells, 2: prominent ballooning cells). MASH was diagnosed when NAS was >5, while non-MASH was diagnosed when NAS was <4. According to the MASH CRN scoring system, a certified pathologist determined the NAS[39]. The histological scoring was performed blindly, without knowledge of any other information.

## Glucose tolerance tests

For the glucose tolerance test, overnight fasted mice were intraperitoneally injected with 2 g/kg D-glucose (Sigma-Aldrich). Glucose concentrations were measured using a Gluco DR. TOP glucometer (Allmedicus). The AUC (area under the curve) during the GTT was calculated using GraphPad Prism 8.0 software (La Jolla, CA, USA) by the trapezoidal method.

## Blood profiling

Blood samples were obtained by cardiac puncture, and plasma was isolated. Blood profiling was performed by GC Pharma (Green Cross

Corporation). Plasma TG and cholesterol were detected with an in vitro enzymatic colorimetric assay (CHOL2 and TRIGL; Roche, Germany) using a Cobas 8000 clinical analyzer with a c702 module (Roche).

## Quantification of hepatic triglyceride
Liver tissues were homogenized in 5% NP-40 using FastPrep-24. To solubilize fat, the homogenates were heated to 95 °C for 5 min and cooled at 23 °C, and repeated. Triglyceride Reagent (Sigma-Aldrich) or DW was added and incubated at 37 °C for 30 min to hydrolyze TG into glycerol. For the colorimetric assay of hydrolyzed TG levels, samples were incubated with Free Glycerol Reagent (Sigma-Aldrich) at 37 °C for 5 min. Differences in absorbance at 540 nm between hydrolyzed or non-hydrolyzed TG were quantified using a glycerol standard (Sigma-Aldrich). TG contents were normalized by the protein concentrations of homogenates, which were measured with a BCA Protein Assay Kit (Thermo Fisher Scientific).

## Quantitative RT-PCR analysis
Total RNA extractions from harvested tissues were performed using TRIzol according to the manufacturer's protocol. After TURBO DNase (Invitrogen) treatment, 2 µg of total RNA was used to generate complementary DNA with a High-Capacity cDNA Reverse Transcription Kit (Thermo Fisher Scientific). Quantitative RT-PCR was performed with Fast SYBR Green Master Mix (Thermo Fisher Scientific) and a Viia 7 Real-time PCR System (Thermo Fisher Scientific) according to the manufacturer's instructions. Gene expression was relatively quantified based on the delta-delta Ct (threshold cycle) method with the beta-actin gene as a reference gene. The sequence of primers can be found in the Supporting Information (Supplementary Table 2).

## Blood−brain barrier (BBB) penetration study in mice
A single oral (gavage) dose of [$^{14}$C]-compound **11c** to male Sprague Dawley rats at 5 mpk and 100 µCi/kg resulted in radioactivity was broadly distributed and detected by quantitative whole-body auto-radiography in most tissues analyzed. At approximately 0.25, 0.5, 1, 6, 24, 48, 96, and 168 h postdose, one Crl:CD(SD) rat/time point was anesthetized by isoflurane inhalation, and a blood sample of approximately 4 mL was collected via cardiac puncture into tubes containing K$_2$EDTA. Following blood collection, each anesthetized animal was euthanized by CO$_2$ inhalation and carcasses were submerged in a dry ice/hexane bath. Frozen carcasses were stored at approximately -20 °C until processed for QWBA. Duplicate aliquots of approximately 0.1 g whole blood were weighed into combustion boats for analysis by LSC. Duplicate aliquots of approximately 0.1 mL plasma were weighed into scintillation vials and mixed with an appropriate volume of scintillation cocktail for analysis by LSC. The frozen carcasses were individually set in a mold, submerged in 5% (w/v) low viscosity carboxymethylcellulose (CMC), and embedded by placing the stage (mold) in a dry ice/hexane bath. Once the blocked carcasses (sample blocks) were frozen, they were removed from the dry ice/hexane bath and placed in storage (approximately -20 °C) for at least 12 h prior to sectioning. Sagittal, 30-µm thick sections of the CMC-embedded carcasses, including QC standards, and of the CMC-embedded calibration standards were sectioned using a Vibratome 9800 Cryomicrotome (Vibratome; St. Louis, MO) with the temperature maintained at approximately -20 °C. The sections were dehydrated for at least 48 h in a frost-free freezer, and then allowed to acclimate to room temperature in a desiccant storage box. The sections were placed in direct contact with the phosphor imaging plates (IP), BAS-SR 2025 (FujiFilm; Tokyo, Japan). The IPs were exposed for approximately 4 days while stored at room temperature in a lead-shielded box to minimize background radiation. Digitized phosphor images (auto-radioluminograms) were generated in a dark room with subdued

lighting using a GE Healthcare Typhoon 7000 Phosphor Imager (Raytest GmbH; Berlin, Germany) and analyzed using AIDA software (Raytest GmbH).

## Pharmacokinetic study in rats and dogs
Compound **11c** was administered to Sprague-Dawley rats and Beagle dogs at 5 mpk via oral gavage (Vehicle: 50 mM Citrate buffer (pH 3.0)) and 5 mpk via intravenous route (Vehicle: NMP + PEG400 + 10% Solutol HS in saline (5:10:50:35, v/v/v/v)). Blood samples were collected at the scheduled times up to 24 h post-dose (IV: 0.083, 0.25, 0.5, 1, 2, 4, 8, 12 and 24 h post-dose, PO: 0.25, 0.5, 1, 2, 4, 8, 12 and 24 h post-dose). Determination of compound **11c** in plasma was performed by UHPLC-MS/MS. Pharmacokinetic parameters were calculated by non-compartmental analysis using Phoenix® WinNonlin® (Ver. 8.3′, Certara).

## Reporting summary
Further information on research design is available in the Nature Portfolio Reporting Summary linked to this article.

# Data availability
All the data supporting the findings of this study are available within the article and its supplementary information files and from the corresponding author. A reporting summary for this article is available as a Supplementary Information file. Source data are provided with this paper.

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

## Acknowledgements

This work was supported by grants from the Ministry of Science, ICT & Future Planning (MSIP)/National Research Foundation of Korea (NRF) (2016M3A9B6902868 (J.H.A), 2021R1A2C2008062 (J.H.A) 2020M3A9E4038695 (H.K)). This work was supported by the Korea Institute of Energy Technology Evaluation and Planning (KETEP) grant funded by the Korea government (MOTIE) (20202020800330, Development and demonstration of energy efficient reaction separation purification process for fine chemical industry) (J.H.A). This research was supported by Korea Drug Development Fund funded by Ministry of Science and ICT, Ministry of Trade, Industry, and Energy, and Ministry of Health and Welfare (RS-2023-00218035, Republic of Korea) (J.H.A). This research was supported by a grant of the Korea Health Technology R&D Project through the Korea Health Industry Development Institute (KHIDI), funded by the Ministry of Health & Welfare, Republic of Korea (grant number: HR22C1832) (I-K.L, J.-H.J).

## Author contributions

I-K.L., H.K. and J.H.A. contributed to the overall design and supervised the study together with R.L, D.K, P.C.G. and S.S. H.S.P., S.H.P., M.K., J.Y. contributed to the design and synthesis of chemicals. B.J., W-I.C., C.J.O. and C.W.L. contributed to the in vivo studies. S.S.K. and M.A.B. contributed to the DMPK studies. J-H.J., C.J.O., E.Y.L. and H.J.L. contributed to the in vitro studies. H.K. and J.H.A. were responsible for drafting the manuscript. All authors have read and revised the article and approved the submitted version.

## Competing interests

A.J.H., H.S.P., S.H.P., M.H.K. and W.M.L. have filed a patent application encompassing aspects of this work (US 2023/0002349 A1 and WO 2021/086133 A1). The remaining authors declare no competing interests.
