## [Peer Review File · Nature Communications]

Discovery of a peripheral 5HT_{2A} antagonist
as a clinical candidate for
Metabolic Dysfunction-Associated SteatohepatitisEditorial Notes:

Parts of this Peer Review File have been redacted as indicated to remove third-party material where no permission to publish could be obtained. Parts of this Peer Review File have been redacted as indicated to maintain the confidentiality of unpublished data.

REVIEWER COMMENTS

Reviewer #1 (Remarks to the Author):

The author studied the 5-HT receptor selectivity and inhibitory effect on animal NAFLD of compound 11c, which may be a 5-HT_{2A} antagonist. The activity of its 5-HT receptor antagonist and its therapeutic effect on NAFLD have been preliminarily determined. The value of the manuscript should be recognized. However, the manuscript is relatively rough, and both the textual description, experimental methods, and conclusions show some errors. The specific manifestations are as follows:

1. In Figure 2, the effect of compound 11c show a significant dose-dependent effect (the effect of 10 mpk is significantly better than that of 5 mpk). However, in Figure 3, the effect of 11c is not dose-dependent, and the efficacy of 5mpk seems to be better than 10mpk. How can this be explained?
2. In "Methods", the author claims to use HEK293 cells (a type of cell derived from kidney embryos) for cell research, however, the data in Table 5 are the results of hepatocyte experiments. What happened?
3. Unreasonable statistical analysis. The score of liver tissue staining in qualitative response data should be evaluated using U-test or chi square test, while other quantitative response data can be evaluated using one-way ANOVA and LSD or other test.
4. In "Historical Analysis", the "NAFLD activity score" method should be briefly explained.
5. In "Glucose Tolerance Tests", explain the calculation method of AUC.
6. Table 5 lists the selective inhibition rate data of 11c on different serotonin receptors. However, there is no specific experimental method in the 'method'. The experimental methods should be supplemented. And the data shows that 11c has good inhibitory effects on 5-HT_{2A}, 2B, 2C, and 5-HT₇. Considering 5-HT_{2C}, 7 is mainly distributed in the central

nervous system, while both 5-HT_{2A} and 5-HT_{2B} are widely distributed in the periphery. Therefore, the conclusion that 11c is a 5-HT₂ antagonist and has the strongest inhibitory effect on 5-HT_{2A} is reasonable, instead of 5-HT_{2A} antagonist.

7. Errors in the text:

Abbreviations in the text and table should be explained, indicating their full name.

Line 56: NAFLD is repeated to indicate abbreviations, unnecessary.

The incorrect Scale bars values are shown in Figures 2e and 2f, where Scale bars=100 mm?

Line 202: What is the method of administration for "oral injection" Is there a conceptual error?

In Figure 3, there is an error in the description of the group. CDAFHD+GM-11c or CDAFHD+11c?

Check the entire text and correct any errors. Such as line spacing, incorrect descriptions, etc.

8. There may be some language grammar and editing errors in the text, and it is recommended that the language is modified by a language polishing service.

Reviewer #2 (Remarks to the Author):

With Focus on the medicinal chemistry work, it seems rather superficially described and illogical. For example, compound 2 has a free amine (ammonium at phys pH) - which must be essential to target binding. Thus there is no rationale for design and synthesis of compounds 1,3-5. The authors should explain why they decided to make these - not just saying modelling suggested. In fact the modelling - docking poses shown - do not highlight any guest-host interactions, nor are backed up by docking of other relevant compounds. Scoring function should also be included.

11b is stated as being metabolically unstable. No data values are given nor any assay. I wonder if the authors have investigated the stability of 11c in buffer/plasma?

11c is chiral - did the authors consider difference in pharmacological/DMPK profile of the two enantiomers?

The choice of R groups seems quite random - and in no way addresses chemical space.

Finally, a 5-HT_{2A} receptor antagonist was previously investigated.

<https://www.sciencedirect.com/science/article/pii/S0024320522010153?via=ihub>

This work was published online Dec 26, 2022 and is not referenced.

All in all, the quality of the manuscript, together with the lack of novelty, I cannot recommend publication in Nat Comm. I suggest a revised manuscript is submitted to a more specialized journal.

Reviewer #3 (Remarks to the Author):

In this paper, the authors based on their previous studies on 5-HT_{2A} LKO mice and 5HT_{2A} antagonists, try to synthesize a serial of new derivatives based on the compound 2 (Clarinet) to develop oral drug candidates of peripheral 5HT_{2A} antagonists and found that compound 11c has the best in vitro inhibitory effect on 5HT_{2A} with a IC₅₀ of 14nM. Then the authors tested the pharmacokinetics of compound 11c both in rats and dogs by iv and gavage administration. Besides, the tissue distribution of compound 11c was also determined by radio labeled [¹⁴C]-11c. After then, the authors explored the anti-NASH effects of compound 11c in vivo on two diet-induced mouse models, and proved compound 11c could significantly improve histologic features of NASH and fibrosis. The following questions may need to be answered by the authors.

1. In general, after gavage administration for 8 weeks (page 178-179) or 12 weeks (page 274-275), the compound 11c mainly distributed in digestive track especially in intestines with comparatively lower plasma drug concentration as well as in the other organs. Do these results indicate the unsatisfactory oral absorption and bioavailability of compound 11c?
2. Did the authors check the 5HT_{2A} inhibitory effects of the compound 11c in vivo? Since this result may establish in vivo connection of its anti-NASH effects with its 5HT_{2A} antagonist role. In in vivo anti-NASH studies, no dose-dependent effects were observed at the doses of 5 and 10 mpk in CDAHFD-induced liver fibrosis and inflammation.
3. Why the authors directly picked the compound 11c (IC₅₀ 14 nM) but without consider the compound 11d? its IC₅₀ (47 nM) is comparable to the compound 11c.
4. Pharmacokinetics studies in rat, why the t_{1/2} of compound 11c was shorter in oral administration (2.5h) than iv injection (4.14h)?
5. In the introduction, 5-HT inhibition or knockdown in liver attenuated lipid deposition and lipid de novo synthesis. How did 5HT_{2A} antagonist inhibit liver inflammation and fibrosis? Did the compound 11c have any effects on lipid metabolism? The plasma TC and TG may also need to be analyzed.

Responses to Reviewers' Comments

We deeply appreciate the constructive comments from the reviewers. The manuscript has been revised following the comments and suggestions of the reviewers. Please find our point-by-point responses to each comment shown below. We hope that all the explanations and additional information can clarify all the concerns raised by the reviewers.

Also, we would like to emphasize that some of the data is only meant for support purpose and should not be accessed by anyone else except for the reviewers. We apologize for not being able to include some of the results in this manuscript. Unfortunately, we are still in the process of publishing the results from STZ+HFD mice along with the results from hepatocyte-specific knockout mice in a separate manuscript. We appreciate your understanding on this matter.

Reviewer #1 (Remarks to the Author):

The author studied the 5-HT receptor selectivity and inhibitory effect on animal NAFLD of compound 11c, which may be a 5HT_{2A} antagonist. The activity of its 5-HT receptor antagonist and its therapeutic effect on NAFLD have been preliminarily determined. The value of the manuscript should be recognized. However, the manuscript is relatively rough, and both the textual description, experimental methods, and conclusions show some errors. The specific manifestations are as follows:

Comment 1: In Figure 2, the effect of compound 11c show a significant dose-dependent effect (the effect of 10 mpk is significantly better than that of 5 mpk). However, in Figure 3, the effect of 11c is not dose-dependent, and the efficacy of 5mpk seems to be better than 10mpk. How can this be explained?

Response: We understand the reviewer's concern about this difference in dose dependency. As described in our manuscript, we evaluated the in vivo efficacy of compound 11c in two different mouse models. Compound 11c showed the in vivo efficacy in a dose-dependent manner in DIO model (original Figure 2), however the effects in choline-deficient, L-amino acid-defined high-fat diet (CDAHFD) model were not dose-dependent (original Figure 3), 5 mpk slightly tends to work better than 10 mpk. The discrepancy between these doses is thought to be primarily attributed to different animal models *per se* or both doses are in error (within error range). Based on these two in vivo results, we first concluded that the efficacy dose of compound 11c in vivo is between 5 – 10 mpk.

We also think that the different mechanisms of fatty liver development and NAFLD progression in these mouse models can result in different dose-dependent responses to compound 11c. In DIO model, the increased hepatic FFA uptake induced fatty liver but was unlikely to induce hepatic fibrosis. In contrast, the increased hepatic uptake of FFA and the defective excretion of FFA from the liver induced severe fatty liver and subsequent liver damage in CDAHFD model. Therefore, we used the DIO model to test the efficacy of compound 11c on the fatty liver and early NASH, and CDAHFD model to test the efficacy of compound 11c on hepatic fibrosis.

5HT signaling through 5HT_{2A} induces lipogenesis in the liver, and compound 11c, an 5HT_{2A} antagonist, can inhibit hepatic lipid accumulation in a dose-dependent manner in the DIO model. However, compound 11c could partially inhibit hepatic lipid accumulation in the CDAHFD model because 5HT did not affect FFA excretion from the liver (reference Nat. Commun. 2018 16;9(1):4824.). Therefore, we did not expect the dose-dependent

improvement of fatty liver in CDAHFD model. We only tested compound 11c in CDAHFD model to see if this drug can prevent or delay the progression of hepatic fibrosis.

Currently, available NAFLD mouse models do not represent the whole spectrum of NAFLD progression. They do not accurately reflect the human NAFLD spectrum, including metabolic dysfunctions and the sequential progression from fatty liver to hepatocellular carcinoma. In this regard, we established a new NAFLD mouse model by modifying STAM (STZ+HFD) mouse model (Reviewer-only Data 1). The previous STAM mouse model, which is administrated with a high dose of STZ (200 ug) at postnatal day 2 and HFD was fed from 4 weeks of age, do not develop obesity and metabolic phenotypes such as insulin resistance, dyslipidemia, and so on (Reviewer-only Data 1A). To overcome this limitation, we established a modified STAM mouse model, in which the HFD was fed continuously after 5 consecutive injections of low-dose STZ (40 mg/kg) at 7 weeks of age (Reviewer-only Data 1B). STZ+HFD mice showed body weight gain (Reviewer-only Data 2A, B), glucose intolerance (Reviewer-only Data 2C, D), insulin resistance (Reviewer-only Data 2E), dyslipidemia, adipocyte hypertrophy, and adipocyte inflammation (Reviewer-only Data 2F). These phenotypes indicated that STZ+HFD mice mimicked metabolic dysfunction like type 2 diabetes accompanied by insulin resistance and beta cell dysfunction (Reviewer-only Data 2).

[figure redacted to maintain the confidentiality of unpublished data]

Reviewer-only Data 1. Several diet-induced NAFLD mouse models and STZ+HFD mice. (A) comparison of diet-induced NAFLD mouse models (Cell Metab. 2019; 29(1):18-26). **(B)** the brief introduction of STZ+HFD mice.

[figure redacted to maintain the confidentiality of unpublished data]

Reviewer-only Data 2. STZ+HFD mice induced obesity and associated metabolic disturbance. (A-F) The 7-week-old B6J mice were treated with STZ and were fed SCD or HFD for 6-68 weeks. **(A)** Body weight trends; $n \geq 6$ per group. **(B)** Fat mass weight of STZ-treated mice; $n = 6-10$ per group. **(C, D)** Intraperitoneal glucose tolerance test (PTGTT) after 14 h fasting with calculated area under the curve (AUC) at 20 weeks of age grouped by diet and STZ treatment; $n = 5$ per group. **(E)** HOMA-IR levels of STZ-treated mice; $n = 5-10$ per group. **(F)** Representative histology by H&E staining of sections from eWAT. Scale bars, 100 μ m.

STZ+HFD mice showed creamy-colored hepatomegaly due to triglyceride accumulation, followed by grossly visible hepatic tumor (Reviewer-only Data 3A). STZ+HFD mice developed grossly visible hepatic tumors from 38 weeks of age, and the incidence and number of the tumors gradually increased to ~90 % and ~1.4 at 56 weeks of age (Reviewer-only Data 3B). Plasma AST/ALT levels were also increased (Data not shown). Furthermore, STZ+HFD mice exhibited sequential histological changes of NAFLD from NAFL (nonalcoholic fatty liver) to NASH (nonalcoholic steatohepatitis), hepatic fibrosis, and HCC (hepatocellular carcinoma) (Reviewer-only data 3C). NAFLD activity score (NAS) gradually increased by 32 weeks of age (Reviewer-only data 3D). Hepatic fibrosis developed from 20 weeks of age and progressed to bridging fibrosis by 32 to 44 weeks of age (Reviewer-only data 3E, F). These data suggested that STZ+HFD mice represented systemic metabolic

disturbances mimicking the physiological changes commonly occurring in human patients (Reviewer-only Data 3).

[figure redacted to maintain the confidentiality of unpublished data]

Reviewer-only Data 3. HFD feeding after streptozotocin treatment induced sequential progression of NAFLD spectrum. (A-F) The 7-week-old B6J mice were treated with STZ and were fed SCD (STZ+SCD) or HFD (STZ+HFD) for 6-68 weeks. (A) Representative liver gross image of STZ-treated mice. Scale bars, 1 cm. (B) Hepatic tumor incidence of STZ+HFD mice; n = 3-10 per group. (C) Representative liver histology by hematoxylin and eosin (H&E) staining. Scale bars, 100 μ m. (D) Non-alcoholic fatty liver disease score (NAS) graded by the NASH CRN scoring system of STZ-treated mice; n = 3-10 per group. (E) Masson's trichrome staining for hepatic fibrosis from STZ-treated mice. Scale bars, 100 μ m. (F) fibrosis stage graded by the METAVIR scoring systems of STZ-treated mice; n = 3-10 per group.

Next, we tested compound 11c in STZ+HFD mice (GM60106 is a clinical candidate, compound 11c in the manuscript). We submitted all these data to the Therapeutic Goods Administration (Australian regulatory authority) for permission for a phase 1 clinical trial and now are in a phase 1 clinical trial. As shown in Reviewer-only data 4, compound 11c (GM60106) effectively improved fibrosis stage and total NAFLD activity score in STZ+HFD mice, and its efficacy was better than obeticholic acid. The property of 5HT_{2A} in metabolic disorders, including obesity and NAFLD, was well characterized (Choi et al., 2018, Nat Com; Song et al., 2020, Endocrinol Metab). Accordingly, we evaluated the potential in vivo efficacy of compound 11c for NASH and NASH-related liver fibrosis in STZ-HFD mice. Mice were intraperitoneally injected with low-dose streptozotocin (40 mg/kg; 5 times) in 7-week-old mice. After injection with STZ, mice were fed an HFD for 16 weeks to induce NASH and liver fibrosis. After 16 weeks of an HFD, mice were fed a standard chow diet to mimic the NASH patient under diet control. At the same time, mice were daily administrated with 5 mg/kg or 10 mg/kg of compound 11c for nine weeks by oral gavage (Reviewer-only data 4).

STZ+HFD mice treated with compound 11c-treated mice showed improved hepatic steatosis, lobular inflammation, hepatocyte ballooning, and total NAFLD activity score by the NASH CRN histological scoring system (Reviewer-only data 4B, C). STZ+HFD mice treated with compound 11c showed an improvement in the hepatic fibrosis stage similar to STZ+HFD mice treated with OCA (Reviewer-only data 4D, E). STZ+HFD mice treated with compound 11c also showed decreased plasma AST and ALT levels, indicating improved liver function (Reviewer-only data 4F). Hepatic triglyceride levels were also reduced in the compound 11c-treated group (Reviewer-only data 4G). Furthermore, the expression of genes related to lipogenesis and fibrosis was reduced in the compound 11c-treated group in a dose-dependent manner (Reviewer-only data 4H, I). These results indicated that compound 11c showed therapeutic potential against NASH and hepatic fibrosis.

[figure redacted to maintain the confidentiality of unpublished data]

Reviewer-only Data 4. Compound 11c treatment with dietary modification ameliorates NASH and liver fibrosis in a dose-dependent manner. (A) Experimental scheme for in vivo efficacy of compound 11c in

(STZ+HFD mice. **(B-I)** STZ+HFD mice fed HFD for 16 weeks were switched to SCD for 9 weeks and were treated with vehicle, compound 11c, or OCA by daily oral gavage. **(B)** Representative liver histology by H&E staining from vehicle, compound 11c 5 mg/kg, compound 11c 10 mg/kg, and OCA (obeticholic acid, reference) 10 mg/kg. Scale bars, 100µm. **(C)** Total NAFLD activity score (NAS) of vehicle, compound 11c, or OCA-treated mice; $n = 9$ per group. **(D)** Representative liver histology by Masson's trichrome staining from vehicle, compound 11c 5 mg/kg, compound 11c 10 mg/kg, and OCA 10 mg/kg. Scale bars, 100µm. **(E)** Fibrosis stage of vehicle, compound 11c, and OCA treated mice; $n = 9$ per group. **(F)** Serum AST and ALT levels; $n = 9$ per group. **(G)** Serum TG levels and hepatic TG levels; $n = 9$ per group. **(H)** Relative mRNA expression of lipogenesis-related genes as assessed by qRT-PCR in the liver; $n = 9$ per group. **(I)** Relative mRNA expression of fibrosis-related genes as assessed by qRT-PCR in the liver; $n = 9$ per group. Data are expressed as the means \pm SEM. * $P < 0.05$, ** $P < 0.01$, one-way ANOVA with post hoc Tukey's test (C, E, F, G, H, I).

Next, we determined the effect of hepatocyte-specific 5HT_{2A} knockout (5HT_{2A} LKO) in STZ+HFD mice. 5HT_{2A} LKO mice were intraperitoneally injected with low-dose streptozotocin (40 mg/kg; 5 times) and fed with HFD for 18, 24, and 32 weeks (mSTAM 26w – NAFL; mSTAM 32W – NASH; mSTAM 40W – late NASH with advanced fibrosis, Reviewer-only data 5A). 5HT_{2A} LKO mice showed improved hepatic steatosis, lobular inflammation, hepatocyte ballooning, and total NAFLD activity score (Reviewer-only data 5B, C). 5HT_{2A} LKO mice also showed amelioration in the hepatic fibrosis stage (Reviewer-only data 5D, E). These data suggested that 5HT_{2A} in hepatocytes could be a therapeutic target against NASH and hepatic fibrosis (Reviewer-only data 5).

[figure redacted to maintain the confidentiality of unpublished data]

Reviewer-only Data 5. 5HT_{2A} deficiency in hepatocytes protects against NASH and liver fibrosis in STZ+HFD mice. (A) The experimental scheme in STZ+HFD mice using 5HT_{2A} hepatocyte-specific knockout mice. (B-F) STZ+HFD mice were fed HFD for 18, 24, and 32 weeks. (B) Representative liver histology by H&E staining. Scale bars, 100µm. (C) Total NAFLD activity score (NAS) in STZ+HFD mice; $n = 8$ per group. (D) Representative liver histology by Masson's trichrome staining and Sirius Red staining. Scale bars, 100µm. (E) Fibrosis stage in STZ+HFD mice; $n = 8$ per group. Data are expressed as the means \pm SEM. * $P < 0.05$, ** $P < 0.01$, one-way ANOVA with post hoc Tukey's test (C, E).

Comment 2: In "Methods", the author claims to use HEK293 cells (a type of cell derived from kidney embryos) for cell research, however, the data in Table 5 are the results of hepatocyte experiments. What happened?

Response: We appreciate the Reviewer's comment. HEK293 cells were used for inhibition assay (cell research), shown in Table 1-3 of the main manuscript, while, hepatic cell lines were used for chemical profiling and for "Hepatocyte stability" test, shown in Table 5 of the main manuscript. The "Selectivity" of Compound 11c for different 5HT subtypes is elucidated in response to Comment 6.

We have modified the Methods section to provide clarity. We have changed the following:

Human 5HT_{2A} Inhibition Assay. The derivatives were evaluated at a concentration of 1 μ M for their antagonist activity on the 5-HT_{2A} receptor (Table 1-3). Human 5-HT_{2A} stable cells were seeded in a 96-well plate (Costar; 3603) at a density of 9×10^4 in 100 μ L of minimum essential medium (Welgene; LM007-86) containing 10% dialyzed fetal bovine serum (Gibco; 30067-334) and 1% penicillin/ streptomycin (Gibco; 15140-122) and kept in a 37 °C incubator for 24 h." (Page 18, line 273)

Comment 3: Unreasonable statistical analysis. The score of liver tissue staining in qualitative response data should be evaluated using U-test or chi square test, while other quantitative response data can be evaluated using one-way ANOVA and LSD or other test.

Response: We agree with the reviewer's comment. We have reanalyzed the data using ANOVA for a comparison of more than two groups and modified the figures and Statistical methods section accordingly. (Page 15 line 208, Page 17 line 248, Page 21 line 382)

Comment 4: In "Historical Analysis", the "NAFLD activity score" method should be briefly explained. (KAIST)

Response: We have added a brief description in Methods to clarify the text as the following:

The total NAFLD activity score (NAS), a widely used measure of grading for steatosis, lobular inflammation, and ballooning was calculated for each sample. NAS ranges from 0-8 and is the sum of unweighted scores for steatosis (0: <5%, 1: 5~33%, 2: 33~66%, 3: >66% fat accumulation in the liver), lobular inflammation (0: no foci inflammation in x200 field, 1: <2 foci inflammation, 2: 2~4 foci inflammation, 3: >4 foci inflammation), and hepatocellular ballooning (0: none, 1: few ballooning cells, 2: prominent ballooning cells). NASH was diagnosed when NAS was >5, while non-NASH was diagnosed when NAS was <4. According to the NASH CRN scoring system, a certified pathologist determined the NAS (Kleiner et al., 2005, Hepatology]. The histological scoring was performed blindly, without knowledge of any other information. (Page 21, line 345)

Comment 5: In "Glucose Tolerance Tests", explain the calculation method of AUC.

Response: The GraphPad Prism 8.0 software computes AUC by the trapezoidal method. It calculates each trapezoid's area, which is described by each connecting segment, by calculating the equivalent rectangle's area. The area under the curve is the sum of the areas of all the rectangles.

The following text was added:

"For the glucose tolerance test (GTT), mice were fasted overnight and intraperitoneally injected with 2 g/kg D-glucose (Sigma-Aldrich). Glucose concentrations were measured using a Gluco DR. TOP glucometer (Allmedicus). The AUC (area under the curve) during the GTT was calculated using GraphPad Prism 8.0 software (La Jolla, CA, USA) by the trapezoidal method". (Page 22, line 355)

Comment 6: Table 5 lists the selective inhibition rate data of 11c on different serotonin receptors. However, there is no specific experimental method in the 'method'. The experimental methods should be supplemented. And the data shows that 11c has good inhibitory effects on 5-HT_{2A}, 2B, 2C, and 5-HT₇. Considering 5-HT_{2C},

7 is mainly distributed in the central nervous system, while both 5-HT2A and 5-HT2B are widely distributed in the periphery. Therefore, the conclusion that 11c is a 5-HT2 antagonist and has the strongest inhibitory effect on 5-HT2A is reasonable, instead of 5-HT2A antagonist.

Response: We appreciate the Reviewer's comment. We have added the text in the method section as the following:

"For further investigation to determine the IC₅₀ values of other 5HT subtypes, Inhibition assay was performed by Eurofins Cerep (Study ID: FR095-0014987, FR095-0016438). The detailed experimental conditions are shown below 11c was tested at several concentrations ranging from 3.0E-09 M to 1.0E-05 M, and the IC₅₀ value was calculated from concentration-response curves. (Page 19, line 284)

In addition we added the table as shown below in methods.

Experimental conditions for the 5HT subtypes

Antagonism assay	Source	Stimulus (Serotonin)	Incubation	Measured Component	Detection Method
5HT _{1A}	human recombinant (BA/F3 cells)	15 nM	RT	intracellular [Ca ²⁺]	Fluorimetry
5HT _{1B}	human recombinant (Hela cells)	100 nM	20 min., 37°C	cAMP	HTRF
5HT _{1D}	rat recombinant (CHO cells)	3 nM	28°C	impedance	Cellular dielectric spectroscopy
5HT _{2A}	human recombinant (HEK-293 cells)	100 nM	30 min., 37°C	IP1	HTRF
5HT _{2B}	human recombinant (CHO cells)	30 nM	30 min., 37°C	IP1	HTRF
5HT _{2C}	human recombinant (HEK-293 cells)	10 nM	30 min., 37°C	IP1	HTRF
5HT _{4E}	human recombinant (CHO cells)	30 nM	30 min., RT	cAMP	HTRF
5HT ₆	human recombinant (CHO cells)	300 nM	30 min., 37°C	cAMP	HTRF
5HT ₇	human recombinant (CHO cells)	300 nM	30 min., 37°C	cAMP	HTRF

Comment 7: Errors in the text:

Abbreviations in the text and table should be explained, indicating their full name.

Line 56: NAFLD is repeated to indicate abbreviations, unnecessary.

Response: We have gone through the text to address the abbreviations as suggested by the Reviewer.

The incorrect Scale bars values are shown in Figures 2e and 2f, where Scale bars=100 mm?

Response: We have corrected the typo as the reviewer pointed out to 100 μm for Fig. 3e and 3f. (Page 16, line 216, 217)

Line 202: What is the method of administration for "oral injection" Is there a conceptual error?

Response: We have corrected the typo the reviewer found from "oral injection" to "oral gavage". (Page 16, Line 226)

In Figure 3, there is an error in the description of the group. CDAHFD+GM-11c or CDAHFD+11c?

Check the entire text and correct any errors. Such as line spacing, incorrect descriptions, etc.

Response: We have corrected CDAHFD + GM-11c to CDAHFD + 11c in Figure 4. We gone through the text for additional errors and corrected them as suggested by the reviewer. (Page 17, line 248)

Comment 8: There may be some language grammar and editing errors in the text, and it is recommended that the language is modified by a language polishing service.

Response: We appreciate the Reviewer's suggestion and the manuscript was reviewed.

Reviewer #2 (Remarks to the Author):

With Focus on the medicinal chemistry work, it seems rather superficially described and illogical.

Comment 1: Compound 2 has a free amine (ammonium at phys pH) - which must be essential to target binding. Thus there is no rationale for design and synthesis of compounds 1,3-5. The authors should explain why they decided to make these - not just saying modelling suggested. In fact the modelling - docking poses shown - do not highlight any guest-host interactions, nor are backed up by docking of other relevant compounds. Scoring function should also be included.

Response: We appreciate the reviewer's comment. In this manuscript, we tried to identify novel, orally available and peripheral 5HT_{2A} antagonists from the FDA approved drug library and rational design from FDA approved CNS drugs. Desloratadine (compound 1), which is an oral histamine 1 antagonist peripheral drug used to treat allergies, was identified as a hit with IC₅₀ value of 232 nM for 5HT_{2A}. Therefore, we thought it is a good starting point to develop an orally available peripheral drug. We optimized compound 1 by introduction of diverse functional groups including carbamate, amide, sulfonamide, urea and alkyl groups in order to understand structure activity relationship as can be seen in Table 1. Based on these results, we further optimized the piperidine moiety with diverse alkyl derivatives. In order to identify best alkyl derivatives, we synthesized diverse derivatives and filed patents (US2023000234 and WO2021086133) with all derivatives (more than 70 compounds). This was added to the manuscript (Page 5, line 98) with Figure 1 as shown below:

Based on docking interactions, it was shown that nitrogen group interacted with Serine-242, which could be critical for binding and important for efficacy. We have added the interaction dotted line between Nitrogen and SER-242. The text was added as the following:

"Based on these interactions, we introduced diverse moieties, but found that the nitrogen group derivatives was critical for efficacy. The modeling was analyzed, the nitrogen of compound 2 showed pocket binding interaction with serine-242, SER-242. In addition, we performed docking of 5HT_{2A} antagonists, Risperidone and Zotepine. We compared binding affinities and it was shown that compound **11c** had a better binding affinity than Compound 2 and commercially available Risperidone and Zotepine, (-9.2, -8.0, -8.4, and -6.3 kcal/mol respectively. In our modeling, Risperidone showed better binding than Zotepine, which could correlate to better efficacy that was previously reported (doi:10.1016/S0140-6736(13)60733-3). This analysis further confirms the ability of Compound 11c as 5HT_{2A} inhibitor" (Page 9, line 143)

Comment 2: 11b is stated as being metabolically unstable. No data values are given nor any assay.

Response: We understand the Reviewer's comment, and we conducted liver microsomal stability of compound 11b. As can be seen below, compound 11b was rapidly metabolized in the liver microsome, so compound 11c was further developed. Compound 11b stability data was added in the manuscript. (Page 8, line 127)

Comment 3: I wonder if the authors have investigated the stability of 11c in buffer/plasma?

Response: We conducted plasma stability with 11c and added it in the manuscript and table. (Table 5, Page 12, line 182)

Comment 4: 11c is chiral - did the authors consider difference in pharmacological/DMPK profile of the two enantiomers?

Response: We separated two diastereomers of compound **11c** by using the reported synthesis procedure by Xu et al Letters in Organic Chemistry, 2014, 11, 470-473, GLSynthesis Inc., USA (synthetic route shown below). Diastereomeric esters mixture was synthesized by (-) camphanoyl chloride treated with 3-(2-chloroethyl)-9-hydroxy-2-methyl-6,7,8,9-tetrahydro-4H-pyrido[1,2-a]pyrimidin-4-one 10c (racemic alcohol) which were separated by silica gel column chromatography and crystallization. It's optical rotations were obtained with a JASCO P-2000 polarimeter to confirmed the diastereomers. Optical rotation of (+)-**10f** and (-)-**10g** is $[\alpha]_D^{24.96} = +63.39^\circ$ (c 0.2, methanol) and $[\alpha]_D^{22.77} = -67.68^\circ$ (c 0.2, methanol) respectively. The two diastereomers separately treated with desloratadine (**2**) to afforded (+)-**22f** $\{[\alpha]_D^{24.5} = +24.57^\circ$ (c 0.2, ethanol)} and (-)-**22g** $\{[\alpha]_D^{24.3} = -27.77^\circ$ (c 0.2, ethanol)} diastereomers. Both diastereomers (+)-**22f** (IC₅₀=11.08 nM) and (-)-**22g** (IC₅₀=10.6 nM) showed similar activity and DMPK profile, therefore, we chose racemic **11c** for further biological evaluation. (Page 8, line 133 and supplementary data file)

Comment 5: The choice of R groups seems quite random - and in no way addresses chemical space.

Response: Thank you for this valuable comment. We synthesized diverse derivatives, which were reported in our patent (US2023000234 and WO2021086133 ect), and only the key (selected) compounds were introduced in this manuscript. Regarding the R group, we were inspired from CNS drugs approved by FDA as can be seen below. We have more data but due to limitation of journal we cited our patents for reference.

[figure redacted to remove third-party material where no permission to publish could be obtained.]

Comment 6: Finally, a 5-HT_{2A} receptor antagonist was previously investigated. <https://www.sciencedirect.com/science/article/pii/S0024320522010153?via=ihub>. This work was published online Dec 26, 2022 and is not referenced.

Response: As per reviewer's comment, we included the reference in the manuscript as reference 23. (Page10, line 150 and Page 27, line 488)

In addition: we would like to explain the character and history of SJT41. As can be seen below, SJT41 was reported as a dual FXR agonist and 5-HTR_{2A} antagonist.

Source: <https://www.sjtmolecular.com> (As of 2023 June)

[figure redacted to remove third-party material where no permission to publish could be

obtained.]

SJT4a showed moderate 5HT_{2a} antagonistic activity with an IC₅₀ value of 1.83 uM (Life Sciences 2023, 314, 121315). Based on website information and patents, SJT4a was originally developed as a FXR agonist. This research group conducted target expansion based on our basic research (Nat Comm 2018) to 5HT_{2a} receptor. Our compound is in Phase 1 clinical trial now, also was developed as 5HT_{2a} antagonist for NASH and NAFLD.

Reviewer #3 (Remarks to the Author):

In this paper, the authors based on their previous studies on 5-HT_{2A} LKO mice and 5HT_{2A} antagonists, try to synthesize a serial of new derivatives based on the compound 2 (Clarinex) to develop oral drug candidates of peripheral 5HT_{2A} antagonists and found that compound 11c has the best in vitro inhibitory effect on 5HT_{2A} with a IC₅₀ of 14nM. Then the authors tested the pharmacokinetics of compound 11c both in rats and dogs by iv and gavage administration. Besides, the tissue distribution of compound 11c was also determined by radio labeled [¹⁴C]-11c. After then, the authors explored the anti-NASH effects of compound 11c in vivo on two diet-induced mouse models, and proved compound 11c could significantly improve histologic features of NASH and fibrosis. The following questions may need to be answered by the authors.

Comment 1: In general, after gavage administration for 8 weeks (page 178-179) or 12 weeks (page 274-275), the compound 11c mainly distributed in digestive track especially in intestines with comparatively lower plasma drug concentration as well as in the other organs. Do these results indicate the unsatisfactory oral absorption and bioavailability of compound 11c?

Response: We appreciate the Reviewer's comment. For most oral drugs, the high level of total radioactivity is often observed in gastrointestinal tract in QWBA (Quantitative Whole Body Autoradiography). In the study of QWBA for compound 11c, tissue distribution was evaluated in multiple time points. Total radioactivities in stomach and small intestine were high at early time points and decreased at late time points. On the other hand, those in the large intestine at early time points were low and increased at late time points, and the peak radio-concentration in the large intestine was much smaller than that in stomach and small intestine. It means that radiolabeled compound 11c was absorbed passing through the gastrointestinal tract. The reason for relatively low plasma concentration compared to some tissues is that absorbed compound 11c is widely distributed into tissues. In the pharmacokinetic study, compound 11c is orally available with the high absolute bioavailability (> 60%) in rats and dogs. Compound 11c has a fairly high volume of distribution and a high bioavailability, which means that it has a high membrane permeability and transmembrane effect in the gastrointestinal tract, which has the most significant impact on bioavailability when administered orally. The high concentration of compound 11c in the digestive system is due to its high membrane permeability and first-pass effect (bioavailability), which leads to high concentrations in the kidneys, liver, small intestine, and colon, which are high-perfusion tissues with high blood flow.

Comment 2: Did the authors check the 5HT_{2A} inhibitory effects of the compound 11c in in vivo? Since this result may establish in vivo connection of its anti-NASH effects with its 5HT_{2A} antagonist role. In in vivo anti-NASH studies, no dose-dependent effects were observed at the doses of 5 and 10 mpk in CDAHFD-induced liver fibrosis and inflammation.

Response: We appreciate reviewer's comments. We did not directly check 5HT_{2A} inhibitory effects of 11c in in vivo. Instead, we conducted in vivo PK with diverse doses (5, 10, 20 mpk) for PK and PD co-relation and found that compound 11c is present at a concentration of more than 20 nM (10 ng/mL) or more up to 8 hrs. As we described in the manuscript, compound 11c showed in vivo efficacy at 5 and 10 mpk and 14 nM of 5HT_{2A} inhibitory activity in in vitro.

[Mice]

Route	IV		PO	
Dose (mg/kg)	5	5	10	20
C0 or Cmax (ng/mL)	468.8	467.4	1115.6	2291.7
Tmax (hr)	-	0.3	0.3	0.3
AUClast (ng·hr/mL)	1160.1	1122.6	2422.3	5098.8
t1/2 (hr)	3.9	1.7	2.2	2.0
Bioavailability (F%)	-	94.1	103.2	108.0
CL (mL/kg)	4170.2	-	-	-
Vdss (mL/kg)	10150.6	-	-	-
Dose ratio		1.0	2.0	4.0
Cmax ratio		1.0	2.4	4.9
AUClast Ratio		1.0	2.2	4.5

We understand the reviewer's concern about this difference in dose dependency. As described in our manuscript, we evaluated the in vivo efficacy of compound 11c in two different mouse models. Compound 11c showed the in vivo efficacy in a dose-dependent manner in DIO model (original Figure 2), however the effects in choline-deficient, L-amino acid-defined high-fat diet (CDAHFD) model were not dose-dependent (original Figure 3), 5 mpk slightly tends to work better than 10 mpk. The discrepancy between these doses is thought to be primarily attributed to different animal models *per se* or both doses are in error (within error range). Based on these two in vivo results, we first concluded that the efficacy dose of compound 11c in vivo is between 5 – 10 mpk.

We also think that the different mechanisms of fatty liver development and NAFLD progression in these mouse models can result in different dose-dependent responses to compound 11c. In DIO model, the increased hepatic FFA uptake induced fatty liver but was unlikely to induce hepatic fibrosis. In contrast, the increased hepatic uptake of FFA and the defective excretion of FFA from the liver induced severe fatty liver and subsequent liver damage in CDAHFD model. Therefore, we used the DIO model to test the efficacy of compound 11c on the fatty liver and early NASH, and CDAHFD model to test the efficacy of compound 11c on hepatic fibrosis.

5HT signaling through 5HT_{2A} induces lipogenesis in the liver, and compound 11c, an 5HT_{2A} antagonist, can inhibit hepatic lipid accumulation in a dose-dependent manner in the DIO model. However, compound 11c could partially inhibit hepatic lipid accumulation in the CDAHFD model because 5HT did not affect FFA excretion from the liver (reference Nat. Commun. 2018 16;9(1):4824.). Therefore, we did not expect the dose-dependent improvement of fatty liver in CDAHFD model. We only tested compound 11c in CDAHFD model to see if this drug can prevent or delay the progression of hepatic fibrosis.

Currently, available NAFLD mouse models do not represent the whole spectrum of NAFLD progression. They do not accurately reflect the human NAFLD spectrum, including metabolic dysfunctions and the sequential progression from fatty liver to hepatocellular carcinoma. In this regard, we established a new NAFLD mouse model by modifying STAM (STZ+HFD) mouse model (Reviewer-only Data 1). The previous STAM mouse model, which is administrated with a high dose of STZ (200 ug) at postnatal day 2 and HFD was fed from 4 weeks of age, do not develop obesity and metabolic phenotypes such as insulin resistance, dyslipidemia, and so on (Reviewer-only Data 1A). To overcome this limitation, we established a modified STAM mouse model, in which the HFD was fed continuously after 5 consecutive injections of low-dose STZ (40 mg/kg) at 7 weeks of age (Reviewer-only Data 1B). STZ+HFD mice showed body weight gain (Reviewer-only Data 2A, B), glucose intolerance (Reviewer-only Data 2C, D), insulin resistance (Reviewer-only Data 2E), dyslipidemia, adipocyte hypertrophy, and adipocyte inflammation (Reviewer-only Data 2F). These phenotypes indicated that STZ+HFD mice mimicked metabolic dysfunction like type 2 diabetes accompanied by insulin resistance and beta cell dysfunction (Reviewer-only Data 2).

[figure redacted to maintain the confidentiality of unpublished data]

Reviewer-only Data 3. Several diet-induced NAFLD mouse models and STZ+HFD mice. (A) comparison of diet-induced NAFLD mouse models (Cell Metab. 2019; 29(1):18-26). (B) the brief introduction of STZ+HFD mice.

[figure redacted to maintain the confidentiality of unpublished data]

Reviewer-only Data 4. STZ+HFD mice induced obesity and associated metabolic disturbance. (A-F) The 7-week-old B6J mice were treated with STZ and were fed SCD or HFD for 6-68 weeks. (A) Body weight trends; $n \geq 6$ per group. (B) Fat mass weight of STZ-treated mice; $n = 6-10$ per group. (C, D) Intraperitoneal glucose tolerance test (PTGTT) after 14 h fasting with calculated area under the curve (AUC) at 20 weeks of age grouped by diet and STZ treatment; $n = 5$ per group. (E) HOMA-IR levels of STZ-treated mice; $n = 5-10$ per group. (F) Representative histology by H&E staining of sections from eWAT. Scale bars, 100 μm .

STZ+HFD mice showed creamy-colored hepatomegaly due to triglyceride accumulation, followed by grossly visible hepatic tumor (Reviewer-only Data 3A). STZ+HFD mice developed grossly visible hepatic tumors from 38 weeks of age, and the incidence and number of the tumors gradually increased to ~90 % and ~1.4 at 56 weeks of age (Reviewer-only Data 3B). Plasma AST/ALT levels were also increased (Data not shown). Furthermore, STZ+HFD mice exhibited sequential histological changes of NAFLD from NAFL (nonalcoholic fatty liver) to NASH (nonalcoholic steatohepatitis), hepatic fibrosis, and HCC (hepatocellular carcinoma) (Reviewer-only data 3C). NAFLD activity score (NAS) gradually increased by 32 weeks of age (Reviewer-only data 3D). Hepatic fibrosis developed from 20 weeks of age and progressed to bridging fibrosis by 32 to 44 weeks of age (Reviewer-only data 3E, F). These data suggested that STZ+HFD mice represented systemic metabolic disturbances mimicking the physiological changes commonly occurring in human patients (Reviewer-only Data 3).

[figure redacted to maintain the confidentiality of unpublished data]

Reviewer-only Data 3. HFD feeding after streptozotocin treatment induced sequential progression of NAFLD spectrum. (A-F) The 7-week-old B6J mice were treated with STZ and were fed SCD (STZ+SCD) or HFD (STZ+HFD) for 6-68 weeks. (A) Representative liver gross image of STZ-treated mice. Scale bars, 1 cm. (B) Hepatic tumor incidence of STZ+HFD mice; n = 3-10 per group. (C) Representative liver histology by hematoxylin and eosin (H&E) staining. Scale bars, 100 μ m. (D) Non-alcoholic fatty liver disease score (NAS) graded by the NASH CRN scoring system of STZ-treated mice; n = 3-10 per group. (E) Masson's trichrome staining for hepatic fibrosis from STZ-treated mice. Scale bars, 100 μ m. (F) fibrosis stage graded by the METAVIR scoring systems of STZ-treated mice; n = 3-10 per group.

Next, we tested compound 11c in STZ+HFD mice (GM60106 is a clinical candidate, compound 11c in the manuscript). We submitted all these data to the Therapeutic Goods Administration (Australian regulatory authority) for permission for a phase 1 clinical trial and now are in a phase 1 clinical trial. As shown in Reviewer-only data 4, compound 11c (GM60106) effectively improved fibrosis stage and total NAFLD activity score in STZ+HFD mice, and its efficacy was better than obeticholic acid. The property of 5HT_{2A} in metabolic disorders, including obesity and NAFLD, was well characterized (Choi et al., 2018, Nat Com; Song et al., 2020, Endocrinol Metab). Accordingly, we evaluated the potential in vivo efficacy of compound 11c for NASH and NASH-related liver fibrosis in STZ-HFD mice. Mice were intraperitoneally injected with low-dose streptozotocin (40 mg/kg; 5 times) in 7-week-old mice. After injection with STZ, mice were fed an HFD for 16 weeks to induce NASH and liver fibrosis. After 16 weeks of an HFD, mice were fed a standard chow diet to mimic the NASH patient under diet control. At the same time, mice were daily administrated with 5 mg/kg or 10 mg/kg of compound 11c for nine weeks by oral gavage (Reviewer-only data 4).

STZ+HFD mice treated with compound 11c-treated mice showed improved hepatic steatosis, lobular inflammation, hepatocyte ballooning, and total NAFLD activity score by the NASH CRN histological scoring system (Reviewer-only data 4B, C). STZ+HFD mice treated with compound 11c showed an improvement in the hepatic fibrosis stage similar to STZ+HFD mice treated with OCA (Reviewer-only data 4D, E). STZ+HFD mice treated with compound 11c also showed decreased plasma AST and ALT levels, indicating improved liver function (Reviewer-only data 4F). Hepatic triglyceride levels were also reduced in the compound 11c-treated group (Reviewer-only data 4G). Furthermore, the expression of genes related to lipogenesis and fibrosis was reduced in the compound 11c-treated group in a dose-dependent manner (Reviewer-only data 4H, I). These results indicated that compound 11c showed therapeutic potential against NASH and hepatic fibrosis.

[figure redacted to maintain the confidentiality of unpublished data]

Reviewer-only Data 4. Compound 11c treatment with dietary modification ameliorates NASH and liver fibrosis in a dose-dependent manner. (A) Experimental scheme for in vivo efficacy of compound 11c in (STZ+HFD mice. (B-I) STZ+HFD mice fed HFD for 16 weeks were switched to SCD for 9 weeks and were treated with vehicle, compound 11c, or OCA by daily oral gavage. (B) Representative liver histology by H&E staining from vehicle, compound 11c 5 mg/kg, compound 11c 10 mg/kg, and OCA (obeticholic acid, reference) 10

mg/kg. Scale bars, 100 μ m. **(C)** Total NAFLD activity score (NAS) of vehicle, compound 11c, or OCA-treated mice; $n = 9$ per group. **(D)** Representative liver histology by Masson's trichrome staining from vehicle, compound 11c 5 mg/kg, compound 11c 10 mg/kg, and OCA 10 mg/kg. Scale bars, 100 μ m. **(E)** Fibrosis stage of vehicle, compound 11c, and OCA treated mice; $n = 9$ per group. **(F)** Serum AST and ALT levels; $n = 9$ per group. **(G)** Serum TG levels and hepatic TG levels; $n = 9$ per group. **(H)** Relative mRNA expression of lipogenesis-related genes as assessed by qRT-PCR in the liver; $n = 9$ per group. **(I)** Relative mRNA expression of fibrosis-related genes as assessed by qRT-PCR in the liver; $n = 9$ per group. Data are expressed as the means \pm SEM. * $P < 0.05$, ** $P < 0.01$, one-way ANOVA with post hoc Tukey's test (C, E, F, G, H, I).

Next, we determined the effect of hepatocyte-specific 5HT_{2A} knockout (5HT_{2A} LKO) in STZ+HFD mice. 5HT_{2A} LKO mice were intraperitoneally injected with low-dose streptozotocin (40 mg/kg; 5 times) and fed with HFD for 18, 24, and 32 weeks (mSTAM 26w – NAFL; mSTAM 32W – NASH; mSTAM 40W – late NASH with advanced fibrosis, Reviewer-only data 5A). 5HT_{2A} LKO mice showed improved hepatic steatosis, lobular inflammation, hepatocyte ballooning, and total NAFLD activity score (Reviewer-only data 5B, C). 5HT_{2A} LKO mice also showed amelioration in the hepatic fibrosis stage (Reviewer-only data 5D, E). These data suggested that 5HT_{2A} in hepatocytes could be a therapeutic potent target against NASH and hepatic fibrosis (Reviewer-only data 5).

[figure redacted to maintain the confidentiality of unpublished data]

Reviewer-only Data 5. 5HT_{2A} deficiency in hepatocytes protects against NASH and liver fibrosis in STZ+HFD mice. (A) The experimental scheme in STZ+HFD mice using 5HT_{2A} hepatocyte-specific knockout mice. (B-F) STZ+HFD mice were fed HFD for 18, 24, and 32 weeks. (B) Representative liver histology by H&E staining. Scale bars, 100 μ m. (C) Total NAFLD activity score (NAS) in STZ+HFD mice; $n = 8$ per group. (D) Representative liver histology by Masson's trichrome staining and Sirius Red staining. Scale bars, 100 μ m. (E) Fibrosis stage in STZ+HFD mice; $n = 8$ per group. Data are expressed as the means \pm SEM. * $P < 0.05$, ** $P < 0.01$, one-way ANOVA with post hoc Tukey's test (C, E).

[figure redacted to maintain the confidentiality of unpublished data]

Reviewer-only Data 5. 5HT_{2A} deficiency in hepatocytes protects against NASH and liver fibrosis in STZ+HFD mice. (A) The experimental scheme in STZ+HFD mice using 5HT_{2A} hepatocyte-specific knockout mice. (B-F) STZ+HFD mice were fed HFD for 18, 24, and 32 weeks. (B) Representative liver histology by H&E staining. Scale bars, 100 μ m. (C) Total NAFLD activity score (NAS) in STZ+HFD mice; $n = 8$ per group. (D) Representative liver histology by Masson's trichrome staining and Sirius Red staining. Scale bars, 100 μ m. (E) Fibrosis stage in STZ+HFD mice; $n = 8$ per group. Data are expressed as the means \pm SEM. * $P < 0.05$, ** $P < 0.01$, one-way ANOVA with post hoc Tukey's test (C, E).

Comment 3: Why the authors directly picked the compound 11c (IC₅₀ 14 nM) but without consider the

compound 11d? its IC50 (47 nM) is comparable to the compound 11c.

Response: We considered all potent compounds for further development. As can be seen below, compound 11d showed lower in vitro activity than 11c and low liver microsomal stability. Therefore we have chosen compound 11c for further development (Page 8, line 127)

Comment 4: Pharmacokinetics studies in rat, why the t_{1/2} of compound 11c was shorter in oral administration (2.5h) than iv injection (4.14h)?

Response: Theoretically, half-lives in the elimination phase are the same between oral and intravenous administrations, if the absorption did not affect the elimination phase.

According to the reviewer's comment, we checked thoroughly PK raw data and the process of PK parameters calculation in rat pharmacokinetic study. PK parameters were calculated using Phoenix WinNonlin and half-life was calculated as $t_{1/2} = 0.693 / K$ (elimination rate constant). K was calculated using the plasma concentrations at least 3 measurable time points in the elimination phase. In raw data, the decrease of the plasma concentration at 24 hr after intravenous administration was slightly low compared to that after oral administration. Therefore, the half-life after intravenous administration was longer than that after oral administration as shown below.

Figure 1. Mean (\pm SD) Plasma Concentrations-Time Profiles of GM-60106 following Single Oral Gavage and Intravenous Administration to Sprague-Dawley Rat at 5, 10 and 20 mg/kg

Since the number of animals per group was small (n=3) in the rat pharmacokinetic study, individual variability can make the difference of the half-lives. Also, the fitting method for slope in the elimination phase was 'Best fit' to calculate the K and the Best fit is the default method of Phoenix WinNonlin and used it to avoid manipulation of the data. We found the number of time points used for fitting the slope was different between

oral and intravenous administrations. This can be one of the reasons for the difference of half-lives.

We additionally included rat (10 mpk) oral PK data in the manuscript and concluded the difference of half-lives between oral and intravenous administration was not pharmacokinetic difference and the difference is thought to be due to individual variability and the number of time points used in calculation of the elimination rate constant.

Based on phase 1 clinical trial, compound 11c has a half-life with more than 10 hrs in SAD (single ascending dose) study.

Comment 5: In the introduction, 5-HT inhibition or knockdown in liver attenuated lipid deposition and lipid de novo synthesis. How did 5HT_{2A} antagonist inhibit liver inflammation and fibrosis? Did the compound 11c have any effects on lipid metabolism? The plasma TC and TG may also need to be analyzed.

Response: Our previous studies showed that the expression of *Htr2a* (5HT_{2A}) was robustly increased by a high-fat diet, and the expression of *SREBP1c* is reduced in hepatocyte-specific *Htr2a* knock-out mice, which in turn reduces the expression of *SREBP1c*-related downstream genes. In addition, RNA-seq results showed that the expression of inflammation and fibrosis-related genes is decreased in hepatocyte-specific *Htr2a* knock-out mice (Choi et al., 2018, Nature Communication). These data suggested that reducing hepatic lipid accumulation could prevent the progression of NASH or hepatic fibrosis in NAFLD.

5HT_{2A} mediates the lipogenic action of gut-derived serotonin in liver. **a.** Relative mRNA expression of indicated HTRs as assessed by qRT-PCR in liver of SCD- and HFD-fed C57BL/6J mice; $n = 4$ per group. **b-k.** The 12-week-old WT and *Htr2a* LKO mice were fed SCD or HFD for 8 weeks. Representative gross liver image (**b**) and liver weight (**c**) of HFD-fed WT and *Htr2a* LKO mice; $n = 6-10$ per group. Scale bar, 1 cm (**b**). **d.** Representative liver histology by H&E staining from HFD-fed WT and *Htr2a* LKO mice. Scale bars, 100µm. **e.** NAS of HFD-fed WT and *Htr2a* LKO mice; $n = 6 - 10$ per group. **f.** Hepatic triglyceride levels; $n = 6 - 10$ per group. **g.** Relative mRNA expression of genes involved in lipogenesis as assessed by qRT-PCR in liver; $n = 5$ per group. **h-j.** Go gene sets related to hepatic steatosis (**h**), inflammation (**j**), and fibrosis (**i**); $n = 3$ per group. **k.** Relative mRNA expression of genes involved in inflammation and fibrosis as assessed by qRT-PCR in liver; $n = 5$ per group. Data are expressed as the means \pm SEM. * $P < 0.05$, ** $P < 0.01$, *** $P < 0.001$, Student's t-test (**a, c, e**)

or one-way ANOVA with post hoc Tukey's test (f, g, k).

As mentioned by the reviewer, it is also anticipated that compound 11c alters lipid metabolism in the liver, similar to the effects observed in mice with hepatocyte-specific 5HT_{2A} knock-out. Therefore, we measured hepatic TG levels in high-fat diet-induced NAFLD mice and found that treatment with compound 11c resulted in a significant reduction in serum triglyceride and hepatic triglyceride levels. Also, we checked the expression of de novo lipogenesis-related genes and found that the expression of Srebp1c, a master regulator gene of de novo lipogenesis, was reduced in the compound 11c treated group (10 mpk). The expression of Fasn, a downstream gene of Srebp1c, was also decreased by compound 11c (10 mpk). We have added these results to Figure 3 as panels g, h (Page 14, Line 204 and Page 15, line 206)

REVIEWER COMMENTS

Reviewer #1 (Remarks to the Author):

The author's answers to most of the questions are feasible, but there are still unsatisfactory answers to the following questions:

1. Not satisfied with the answer to 'Comment 2 (In "Methods", the author claims to use HEK293 cells (a type of cell derived from kidney embryos) for cell research, however, the data in Table 5 are the results of hepatocyte experiments. What happened?)'. Moreover, it is even more unclear which cell line was used in the cell experiments in the "method". And, the original cell line has been deleted, what does that mean?

In fact, multiple cell lines were used in cell experiments. Their cultivation methods should be explained separately in the "methods" section, and the respective experiments should be indicated.

2. The statistical analysis (Comment 3: Unreasonable statistical analysis. The score of liver tissue staining in qualitative response data should be evaluated using U-test or chi square test, while other quantitative response data can be evaluated using one-way ANOVA and LSD or other test.) is still unclear. It should be clearly stated in the 'statistical analysis' that which data was analyzed using what statistical analysis, and which other statistical analysis was used for the other data.

In addition, homogeneity of variance should be tested during analysis of variance. There should be differences in the testing methods used when the variance is uniform and uneven. Moreover, it is not appropriate to use Tukey's Honey Significant 330 Difference (HSD) test to analyze discontinuous data (Figure 3f), and U-test and other testing methods should be applied.

3. "p<0.05" or "P<0.05"? Unified representation is required.

Reviewer #2 (Remarks to the Author):

I will repeat my recommendation: No matter changes made (I did not review these) - I cannot recommend publication in Nat Comm due to lack of novelty. I suggest a revised manuscript is submitted to a more specialized journal.

Reviewer #4 (Remarks to the Author):

- 1) I recommend that the authors consider updating the terminology from "Nonalcoholic fatty liver disease (NAFLD)" to "metabolic dysfunction-associated steatotic liver disease (MASLD)" in alignment with the recent multi-society Delphi consensus statement. Please refer to PMID: 37364816, 37364790, and 37363821 for more details.
- 2) While the authors have acknowledged potential conflicts of interest due to a patent application, it's noteworthy that several of them are affiliated with JD Bioscience Inc., the organization overseeing the GM-60106 trial. It would be beneficial to elucidate the specific roles and contributions of these authors in the study design and interpretation of results for clarity.
- 3) The authors have employed two distinct DIO models to assess the efficacy of compound 11c - the HDF model for fatty liver and early NASH and the CDAHFD model for hepatic fibrosis. While these models are widely utilized in NAFLD research, there is also an understanding of the inherent differences in etiology that influence the manifestation of the respective phenotypes. To this end, it would be prudent to include a section discussing the limitations of these models in the Discussion section of the paper.
- 4) In response to Comment 5 from Reviewer 3, the question of how the 5HT2A antagonist mitigates liver inflammation and fibrosis remains partially unanswered. While the authors allude to their previous work, highlighting reduced inflammation and fibrosis gene expression in hepatocyte-specific Htr2a KO mice, this doesn't directly correlate to the systemic effects of the drug in question. The discussion emphasizes altered lipid metabolism in the liver but falls short in addressing inflammation and fibrosis mechanisms. It might be helpful for the authors to specify in the limitations section that while the 5HT2A antagonist has demonstrated effects on liver lipid metabolism potentially preventing the progression of NASH or hepatic fibrosis in NAFLD, the exact mechanisms, particularly related to inflammation and fibrosis, remain uncertain or are currently under investigation.

Reviewer #5 (Remarks to the Author):

Starting from the results of previous research by the same group showing that liver 5HT2A knockout mice suppressed steatosis and reduced fibrosis-related gene expression, in this

manuscript Pagire and colleagues describe the discovery of novel peripheral 5HT2A antagonists and test the effects of the most promising one (compound 11c) in two in vivo models of NAFLD. The final version of the manuscript, which incorporates the numerous and appropriate revisions suggested by the reviewers, clearly expounds the methodology used to develop the research project. The results are impressive and well commented.

In particular, a better explanation of the logic behind the synthesis of compounds was needed.

I believe that, in its current version, the manuscript is publishable.

REVIEWER COMMENTS

Reviewer #1 (Remarks to the Author):

The author's answers to most of the questions are feasible, but there are still unsatisfactory answers to the following questions:

1. Not satisfied with the answer to 'Comment 2 (In "Methods", the author claims to use HEK293 cells (a type of cell derived from kidney embryos) for cell research, however, the data in Table 5 are the results of hepatocyte experiments. What happened?)'. Moreover, it is even more unclear which cell line was used in the cell experiments in the "method". And, the original cell line has been deleted, what does that mean?

In fact, multiple cell lines were used in cell experiments. Their cultivation methods should be explained separately in the "methods" section, and the respective experiments should be indicated.

Answers)

We appreciate the Reviewer's comment. The reason for using HEK293 cells is as follows: The HEK293 cell line (a type of cell derived from kidney embryos), in conjunction with CHO (chinese hamster ovary) cells, is well-suited for genetic modification and gene expression, making it a widely used cell line for manipulating and controlling specific genes or proteins in drug screening research. In this study, we conducted screening for 5HT2A antagonists using 5HT2A stable cells. I have made the correction from 'Human 5HT2A stable cells' to 'Human 5HT2A stably transfected HEK293 cells (a type of cell derived from kidney embryos)' on page 20, line 309 of the manuscript.

Hepatocyte stability test is an experiment used to study the degradation of chemical compounds. In this study, we performed hepatocyte stability for compound 11c following a 3-hour in vitro incubation in human, dog, monkey, mouse, and rat hepatocytes, following Fidelta's protocol (Zagreb, Croatia). The HPLC-HRMS experiments for metabolite identification analysis of compound 11c were carried out using high-resolution mass spectrometry (HRMS) with electrospray ionization in positive and negative modes under basic chromatographic conditions. Positive ionization mode was selected for metabolite identification.

We have revised the method related to the hepatocyte stability assay based on the reviewer's comments. The changes have been incorporated on page 21, line 319 of the manuscript.

Hepatocyte stability test

Hepatocyte incubations for the test compound (10 μ M final concentration, n=2 per each time point) were carried out in mouse, rat, dog, monkey and human cryopreserved hepatocytes, resuspended at a density of 0.5 million cells/ml of modified Krebs-Henseleit buffer (KHB). The information regarding the hepatocytes used in the experiment is as follows: liverPool® 10-donor mixed gender human (BioIVT, Cat.No. X008001, lot EAV), male cynomolgus monkey (BioIVT, Cat.No. M00305, lot COJ), male Beagle dog (BioIVT, Cat.No. M00205, lot USY), male Sprague Dawley rat (BioIVT, Cat.No. M00005, lot TQE) and male CD1 mouse (BioIVT, Cat.No.005052,

lot LFF). In addition, the compound was also incubated with blank KHB. The final concentration of solvent (DMSO) was 0.1% of the final volume. The incubations were performed for 3 hours at 37°C and pH 7.4. Reactions were terminated at 0 and 180 minutes by addition of 3 volumes of stop solution (acetonitrile/methanol (2:1, v/v) containing internal standard–diclofenac at 100 ng/ml). Samples (compound spiked buffer and compound spiked hepatocytes, as well as blank samples, i.e. hepatocytes with same DMSO%) were then centrifuged and aliquots were transferred to 96-deep micronic tubes and frozen at -80°C until LC-MS/MS analysis. The metabolic activity of hepatocytes was verified by simultaneous incubation of following assay controls: testosterone (substrate for Phase I enzymes from CYP3A enzyme family, high clearance in all species), umbelliferone (substrate for Phase II conjugation enzymes: UDP glucuronyl transferases (UGT) and sulfotransferases (SULT); high clearance in all species) and caffeine (stable compound). All controls were within acceptable range in all tested species. The metabolite identification for GM-60106 was done on a Q-TOF X500R high resolution mass spectrometer (Sciex) equipped with a TurboIonSpray source and coupled to Shimadzu Nexera Series 30 HPLC pumps and autosampler. The most abundant detected metabolites are reported with molecular mass, molecular formula, proposed structure, retention time relative to the parent (RRT) and most abundant fragments detected.

2. The statistical analysis (Comment 3: Unreasonable statistical analysis. The score of liver tissue staining in qualitative response data should be evaluated using U-test or chi square test, while other quantitative response data can be evaluated using one-way ANOVA and LSD or other test.) is still unclear. It should be clearly stated in the 'statistical analysis' that which data was analyzed using what statistical analysis, and which other statistical analysis was used for the other data. In addition, homogeneity of variance should be tested during analysis of variance. There should be differences in the testing methods used when the variance is uniform and uneven. Moreover, it is not appropriate to use Tukey's Honey Significant 330 Difference (HSD) test to analyze discontinuous data (Figure 3f), and U-test and other testing methods should be applied.

Answers)

We appreciate the reviewer's comment. Initially, we performed statistical analyses using one-way analysis of variance (ANOVA) to compare more than three groups, followed by Dunnett's test or Tukey's Honestly Significant Difference (HSD) test, as appropriate. A p-value <0.05 was considered to be statically significant (GraphPad Prism 8, MA, USA). Before the *post hoc* ANOVA test, we checked that F achieves the necessary level of statistical significance (P<0.05) and that there is no significant variance in inhomogeneity.

In this revision, as the reviewer suggested, we used the Mann-Whitney *U*-test to assess the statistical significance of the two groups. P values were below 0.05 In all data, which indicates significant differences between each group (Vehicle vs 11c compound 5mpk or Vehicle versus 11c compound 10 mpk) (Figure 1).

We have also corrected the material and method in this manuscript. (Page 24, line 387)

Figure 1. The result of total NAFLD activity score by statistical analysis using Mann-Whitney *U*-test.

3. “ $p < 0.05$ ” or “ $P < 0.05$ ”? Unified representation is required.

Answers)

We appreciate the reviewer’s comment. We expressed it in a uniform notation ‘ $P < 0.05$ ’. (Page 19, line 278)

Reviewer #2 (Remarks to the Author):

I will repeat my recommendation: No matter the changes made (I did not review these) - I cannot recommend publication in Nat Comm due to lack of novelty. I suggest a revised manuscript is submitted to a more specialized journal.

Reviewer #4 (Remarks to the Author):

1) I recommend that the authors consider updating the terminology from "Nonalcoholic fatty liver disease (NAFLD)" to "metabolic dysfunction-associated steatotic liver disease (MASLD)" in alignment with the recent multi-society Delphi consensus statement. Please refer to PMID: 37364816, 37364790, and 37363821 for more details.

Answers)

We appreciate the reviewer's comments. We have changed the terminology from 'NAFLD' to 'MASLD' in this manuscript.

2) While the authors have acknowledged potential conflicts of interest due to a patent application, it's noteworthy that several of them are affiliated with JD Bioscience Inc., the organization overseeing the GM-60106 trial. It would be beneficial to elucidate the specific roles and contributions of these authors in the study design and interpretation of results for clarity.

Answers)

As per reviewer's comments, we have included "Author contributions" in the text

I-K.L., H.K., and J.H.A. contributed to the overall design and supervised the study together with R.L, D.K, P.G., and S.S. H.S.P., S.H.P., M.K., J.Y. contributed to the design and synthesis of chemicals. B.J., W-I.C., C.J.O., C.W.L. contributed to the in vivo studies. S.S.K., and M. A. B. contributed to the DMPK studies. J-H. J., C.J.O., E.Y.L., H.J.L. contributed to the in vitro studies. H.K., and J.H.A. were responsible for drafting the manuscript. All authors have read and revised the article and approved the submitted version. (Page 33, line 621)

3) The authors have employed two distinct DIO models to assess the efficacy of compound 11c - the HDF model for fatty liver and early NASH and the CDAHFD model for hepatic fibrosis. While these models are widely utilized in NAFLD research, there is also an understanding of the inherent differences in etiology that influence the manifestation of the respective phenotypes. To this end, it would be prudent to include a section discussing the limitations of these models in the Discussion section of the paper.

Answers)

As the reviewer suggested, we have included following paragraph in the result and discussion section.

Our previous report also revealed that genetic deletion of 5HT_{2A} resulted in the downregulation of genes related to hepatic fibrosis and inflammation.^{14, 27} (Page 17, Line 236-238)

These beneficial effects of compound 11c on hepatic inflammation and fibrosis could be attributed to the reduction in hepatic lipid accumulation. However, considering that elevated ROS production and mitochondrial stress are important in the development of hepatic inflammation and fibrosis in CDAHFD mouse model, our data also suggested that anti-inflammation and fibrosis effects of compound 11C could be attributed to the inhibition of 5HT_{2A} signaling in other target cells such as macrophage and hepatic stellate cells. (Page 18, line 261-267)

4) In response to Comment 5 from Reviewer 3, the question of how the 5HT_{2A} antagonist mitigates liver inflammation and fibrosis remains partially unanswered. While the authors allude to their previous work, highlighting reduced inflammation and fibrosis gene expression in hepatocyte-specific Htr2a KO mice, this doesn't directly correlate to the systemic effects of the drug in question. The discussion emphasizes altered lipid metabolism in the liver but falls short in addressing inflammation and fibrosis mechanisms. It might be helpful for the authors to specify in the limitations section that while the 5HT_{2A} antagonist has demonstrated effects on liver lipid metabolism potentially preventing the progression of NASH or hepatic fibrosis in NAFLD, the exact mechanisms, particularly related to inflammation and fibrosis, remain uncertain or are currently under investigation.

Answer)

We agree to the reviewer's comments and discuss them in our revised manuscript as below.

We also showed possibility of the direct anti-inflammatory and anti-fibrotic effects of compound 11c using CDAHFD mouse model. Although we did not show any direct evidence on 5-HT's role in inflammation and fibrosis, 5-HT is known to regulate the activation and proliferation of hepatic stellate cells (HSCs) and macrophages (Mederacke et al., 2013)(Ebrahimkhani et al. 2011) (Ruddell et al. 2006, Kim et al. 2013). Thus, further study to elucidate the detailed mechanisms how 5HT_{2A} antagonist can reduce hepatic inflammation and fibrosis is currently under investigation. (Page 20, 295)

Reviewer #5 (Remarks to the Author):

Starting from the results of previous research by the same group showing that liver 5HT_{2A} knockout mice suppressed steatosis and reduced fibrosis-related gene expression, in this manuscript Pagire and colleagues describe the discovery of novel peripheral 5HT_{2A} antagonists and test the effects of the most promising one (compound 11c) in two in vivo models of NAFLD. The final version of the manuscript, which incorporates the numerous and appropriate revisions suggested by the reviewers, clearly expounds the methodology used to develop the research project. The results are impressive and well commented.

In particular, a better explanation of the logic behind the synthesis of compounds was needed. I believe that, in its current version, the manuscript is publishable.

→ We appreciate the reviewer's comments and positive feedback. We are glad that the addition of Fig. 1. Strategy and history of 5HT_{2A} antagonists helped explain the logic behind the synthesis of the compounds.

REVIEWERS' COMMENTS

Reviewer #1 (Remarks to the Author):

Reviewers acknowledge the author's modifications to the article.

Reviewer #5 (Remarks to the Author):

Dear Authors,

I consider the manuscript publishable in its current version.

I have no further suggestions